# Deriving density-matrix functionals for excited states

Julia Liebert[1, 2] and Christian Schilling[1, 2, *]

[1]*Department of Physics, Arnold Sommerfeld Center for Theoretical Physics,*
*Ludwig-Maximilians-Universität München, Theresienstrasse 37, 80333 München, Germany*
[2]*Munich Center for Quantum Science and Technology (MCQST), Schellingstrasse 4, 80799 München, Germany*

(Dated: May 22, 2023)

We initiate the recently proposed $\boldsymbol{w}$-ensemble one-particle reduced density matrix functional theory ($\boldsymbol{w}$-RDMFT) by deriving the first functional approximations and illustrate how excitation energies can be calculated in practice. For this endeavour, we first study the symmetric Hubbard dimer, constituting the building block of the Hubbard model, for which we execute the Levy-Lieb constrained search. Second, due to the particular suitability of $\boldsymbol{w}$-RDMFT for describing Bose-Einstein condensates, we demonstrate three conceptually different approaches for deriving the universal functional in a homogeneous Bose gas for arbitrary pair interaction in the Bogoliubov regime. Remarkably, in both systems the gradient of the functional is found to diverge repulsively at the boundary of the functional's domain, extending the recently discovered Bose-Einstein condensation force to excited states. Our findings highlight the physical relevance of the generalized exclusion principle for fermionic and bosonic mixed states and the curse of universality in functional theories.

## I. INTRODUCTION

The Penrose and Onsager criterion [1] identifies one-body reduced density matrix functional theory (RDMFT) as a potentially ideal approach to describe Bose-Einstein condensation (BEC). Indeed, BEC is present whenever one eigenvalue of the one-particle reduced density matrix (1RDM) is proportional to the total particle number $N$. The 1RDM in turn is the natural variable in RDMFT, which is, at least in-principle, an exact approach to describe interacting $N$-particle quantum systems. The absence of complete condensation for interacting bosons is strongly tied to the concept of quantum depletion which characterizes the fraction of bosons outside the BEC ground state [2]. It has been one of the recent achievements of RDMFT [3, 4] to provide a *universal* explanation for quantum depletion, independently of the microscopic details of the system: The distinctive shape of the universal functional reveals the existence of a BEC force which explains from a purely geometric point of view why not all bosons can condense.

This and other recent progress in the field of ground state RDMFT for bosons [3–7] and $\boldsymbol{w}$-ensemble RDMFT for excited states [8–10] urges us to systematically derive in this paper the first functionals for $\boldsymbol{w}$-ensemble RDMFT. To initiate the development of $\boldsymbol{w}$-ensemble functionals in different fields of physics we derive analytically the universal functional for both the symmetric Hubbard dimer with on-site interaction and the homogeneous Bose gas in the Bogoliubov regime. The latter functional constitutes the bosonic analogue of the Hartree-Fock functional for fermions [11]. Both systems do not only allow us to obtain an analytic expression for the universal functional but are also well-suited to illustrate conceptually different routes for their derivation. Besides illustrating the application of $\boldsymbol{w}$-ensemble

RDMFT for the first time, we extend the concept of a BEC force based on the diverging gradient of the functional close to the boundary of its domain to excited state RDMFT.

The paper is structured as follows. To keep our work self-contained, we recall in Sec. II the basic formalism of $\boldsymbol{w}$-ensemble RDMFT. We illustrate $\boldsymbol{w}$-ensemble RDMFT and derive the exact universal functionals for the symmetric Hubbard dimer in Sec. III and the homogeneous BECs in Sec. IV.

## II. RECAP OF ENSEMBLE RDMFT FOR NEUTRAL EXCITATIONS

Before deriving the first $\boldsymbol{w}$-ensemble functionals in Secs. III and IV we introduce in this section the required foundational concepts of $\boldsymbol{w}$-ensemble RDMFT which has recently been proposed by us for bosons in Ref. [10] and for fermions in Ref. [8, 9]. From a general perspective, RDMFT is based on the observation that in each field of physics the interaction $W$ between the particles is usually kept fixed. As a consequence, one considers all Hamiltonians $\hat{H}$ on the $D$-dimensional $N$-boson Hilbert space $\mathcal{H}_N$ that are parameterized by the one-particle Hamiltonian $\hat{h}$,

$$\hat{H}(\hat{h}) \equiv \hat{h} + \hat{W} . \tag{1}$$

To arrive at a corresponding functional theory, the ensemble RDMFT for excited states combines a variational principle proposed by Gross, Oliveira and Kohn (GOK) [12–14] with the Levy-Lieb constrained search [15, 16]. In the GOK variational principle, the weighted sum $E_{\boldsymbol{w}} \equiv \sum_j w_j E_j$ of the increasingly ordered eigenenergies $E_i$, $E_1 \leq E_2 \leq \ldots \leq E_D$, of the Hamiltonian $\hat{H}$ and decreasingly ordered weights $w_1 \geq w_2 \geq \ldots \geq w_D$ with $\sum_{i=1}^D w_i = 1$ follows from minimizing the energy $\mathrm{Tr}_N[\hat{\Gamma}\hat{H}]$ over all $N$-boson/fermion density operators

* c.schilling@physik.uni-muenchen.de

arXiv:2210.00964v3 [cond-mat.quant-gas] 19 May 2023

with spectrum given by the weight vector, $\text{spec}^\downarrow(\hat{\Gamma}) = \boldsymbol{w}$. This spectral condition defines the set $\mathcal{E}^N(\boldsymbol{w})$ of $N$-particle density operators

$$\mathcal{E}^N(\boldsymbol{w}) \equiv \{\hat{\Gamma} \,|\, \hat{\Gamma} = \hat{\Gamma}^\dagger, \hat{\Gamma} \geq 0, \text{Tr}_N[\hat{\Gamma}] = 1, \text{spec}^\downarrow(\hat{\Gamma}) = \boldsymbol{w}\}. \tag{2}$$

Then, the GOK variational principle reads [12]

$$E_{\boldsymbol{w}} \equiv \sum_{j=1}^{D} w_j E_j = \min_{\hat{\Gamma} \in \mathcal{E}^N(\boldsymbol{w})} \text{Tr}_N\left[\hat{\Gamma}\hat{H}\right]. \tag{3}$$

Applying the Levy-Lieb constrained search [15, 16] to this variational principle for excited states yields

$$\begin{aligned} E_{\boldsymbol{w}}(\hat{h}) &= \min_{\hat{\Gamma} \in \mathcal{E}^N(\boldsymbol{w})} \text{Tr}_N[(\hat{h} + \hat{W})\hat{\Gamma}] \\ &= \min_{\hat{\gamma} \in \mathcal{E}_N^1(\boldsymbol{w})} \left[ \min_{\mathcal{E}^N(\boldsymbol{w}) \ni \hat{\Gamma} \mapsto \hat{\gamma}} \text{Tr}_N[(\hat{h} + \hat{W})\hat{\Gamma}] \right] \\ &= \min_{\hat{\gamma} \in \mathcal{E}_N^1(\boldsymbol{w})} \left[ \text{Tr}_1[\hat{h}\hat{\gamma}] + \mathcal{F}_{\boldsymbol{w}}(\hat{\gamma}) \right], \end{aligned} \tag{4}$$

where we defined the universal functional $\mathcal{F}_{\boldsymbol{w}}$ whose domain is given by $\mathcal{E}_N^1(\boldsymbol{w}) = N\text{Tr}_{N-1}(\mathcal{E}^N(\boldsymbol{w}))$. For simplicity we used in Eq. (3) the same symbol for the one-particle Hamiltonian $\hat{h}$ on the $N$-particle and the one-particle Hilbert space. It is worth stressing here that the set $\mathcal{E}_N^1(\boldsymbol{w})$ is typically not convex [17, 18]. Moreover, $\mathcal{F}_{\boldsymbol{w}}$ is usually not (locally) convex, i.e., there exist convex regions on which $\mathcal{F}_{\boldsymbol{w}}$ is not convex, even for those special cases with convex domain $\mathcal{E}_N^1(\boldsymbol{w})$. A well-known example for the latter scenario is the ground state Hubbard dimer functional for the singlet spin sector which is recovered for the weight vector $\boldsymbol{w} = (1, 0, \ldots)$ [6, 19, 20].

One of the main achievements of Refs. [8–10, 18] was to overcome the too intricate $\boldsymbol{w}$-ensemble $N$-representability constraints that define the domain $\mathcal{E}_N^1(\boldsymbol{w})$ of the universal functional $\mathcal{F}_{\boldsymbol{w}}$. In analogy to Valone's ground state RDMFT [21] this was achieved by performing an exact convex relaxation. Indeed the energy $E_{\boldsymbol{w}}(\hat{h})$ remains unaffected by replacing the non-convex sets $\mathcal{E}^N(\boldsymbol{w})$ and $\mathcal{E}_N^1(\boldsymbol{w})$ by their respective convex hulls,

$$\begin{aligned} \overline{\mathcal{E}}^N(\boldsymbol{w}) &\equiv \text{conv}\left(\mathcal{E}^N(\boldsymbol{w})\right), \\ \overline{\mathcal{E}}_N^1(\boldsymbol{w}) &\equiv N\text{Tr}_{N-1}\left(\overline{\mathcal{E}}^N(\boldsymbol{w})\right) = \text{conv}\left(\mathcal{E}_N^1(\boldsymbol{w})\right). \end{aligned} \tag{5}$$

In particular, inserting Eq. (5) in the Levy-Lieb constrained search replaces $\mathcal{F}_{\boldsymbol{w}}$ by its lower convex envelope

$$\begin{aligned} \overline{\mathcal{F}}_{\boldsymbol{w}}(\hat{\gamma}) &\equiv \min_{\overline{\mathcal{E}}^N(\boldsymbol{w}) \ni \hat{\Gamma} \mapsto \hat{\gamma}} \text{Tr}_N[\hat{\Gamma}\hat{W}] \\ &= \text{conv}\left(\mathcal{F}_{\boldsymbol{w}}(\hat{\gamma})\right). \end{aligned} \tag{6}$$

It was exactly this convex relaxation which allowed us in Refs. [8–10] to obtain a feasible functional theory thanks to a comprehensive characterization of the set $\overline{\mathcal{E}}_N^1(\boldsymbol{w})$ for bosons and fermions. To be more specific, we derived a compact description of the corresponding spectral set

$$\Sigma(\boldsymbol{w}) \equiv \text{spec}(\overline{\mathcal{E}}_N^1(\boldsymbol{w})), \tag{7}$$

in terms of finitely many linear constraints [8–10, 18]. Those conditions represent nothing else than a complete generalization of Pauli's exclusion principle to mixed states of bosons and fermions, respectively. Therefore, the challenging task addressed in this paper is to derive the first $\boldsymbol{w}$-ensemble functionals. This should initiate the development of more elaborated functional approximations in analogy to the developments of ground state RDMFT functionals for fermions [22–34] which were inspired by or even based on the Hartree-Fock functional introduced in the seminal work by Lieb [11].

## III. DERIVATION OF THE UNIVERSAL FUNCTIONAL FOR THE SYMMETRIC BOSE-HUBBARD DIMER

As our first proof-of-principle for $\boldsymbol{w}$-ensemble RDMFT, we derive in this section the exact $\boldsymbol{w}$-ensemble functional for the symmetric Bose-Hubbard dimer. Due to the equivalence of this system to the Fermi-Hubbard dimer for two electrons in their singlet sector, the corresponding results can be translated to fermionic $\boldsymbol{w}$-ensemble RDMFT in a straightforward manner. Understanding the $\boldsymbol{w}$-ensemble functional and its domain for this model is particularly interesting since the Bose-Hubbard dimer constitutes the building block of the Hubbard model widely used in the field of ultracold quantum gases. Besides this, the Hubbard dimer model is widely used throughout RDMFT and density functional theory to illustrate conceptual aspects of functional theory [19, 20, 35–50]. The Hamiltonian for spinless bosons on two lattice sites reads

$$\hat{H} = -t\left(\hat{a}_L^\dagger \hat{a}_R + \hat{a}_R^\dagger \hat{a}_L\right) + U \sum_{j=L,R} \hat{n}_j\left(\hat{n}_j - 1\right), \tag{8}$$

where the first term describes hopping at a rate $t$ between the left $(L)$ and right $(R)$ lattice site. The second term describes the Hubbard on-site interaction with coupling strength $U$ and $\hat{n}_j = \hat{a}_j^\dagger \hat{a}_j$ is the occupation number operator.

In the case of periodic boundary conditions, the Hamiltonian in Eq. (8) is translationally invariant. This implies that the total momentum $P$ is conserved, i.e., $P$ is a good quantum number. As a result, the minimization in the constrained search formalism in Eq. (4) can be restricted to all $\hat{\Gamma} \in \mathcal{E}^N(\boldsymbol{w}, P)$ in the chosen symmetry sector with fixed $P$. Then, it is possible to establish a separate functional in each symmetry sector instead of a single more involved functional referring to all $P$. Moreover, every 1RDM $\hat{\gamma}$ is diagonal in momentum representation. In the following, we consider the case of $N = 2$ spinless bosons and restrict to repulsive interactions, i.e., $U > 0$. Then, the natural occupation numbers are given by the momentum occupation numbers $n_p \geq 0$ restricted through the normalization $\sum_p n_p = 2$.

It is also worth noticing that for the symmetric Bose-Hubbard dimer the translational invariance is equivalent

to the inversion symmetry. To explain this, we now skip the periodic boundary conditions and instead restrict to the even symmetry sector. The corresponding symmetry-adapted one-boson basis consists of the two states $|e\rangle = (|L\rangle + |R\rangle)/\sqrt{2}$ and $|o\rangle = (|L\rangle - |R\rangle)/\sqrt{2}$ which actually coincide with the two one-particle momentum states. The two-dimensional subspace with even inversion-symmetry is then spanned by the two basis states $|e,e\rangle$ and $|o,o\rangle$. This implies that the 1RDM $\hat{\gamma}$ is diagonal and thus depends on only one free parameter $n_e$, the occupation number of $|e\rangle$. Thus, the resulting $\boldsymbol{w}$-ensemble functional $\mathcal{F}_{\boldsymbol{w}}$ is equivalent to $\mathcal{F}_{\boldsymbol{w}}$ in the symmetric Bose-Hubbard dimer with periodic boundary conditions and restricted to $P = 0$ in (10) with $n_0$ replaced by $n_e$.

In the following, we consider the $P = 0$ momentum sector. For two lattice sites, the single particle momentum can take the discrete values $p_\nu = \pi\nu$ with $\nu = 0, 1$. We denote the creation (annihilation) operator referring to the momentum $\nu$ by $\hat{a}_\nu^\dagger$ ($\hat{a}_\nu$) and the occupation number operators by $\hat{n}_\nu = \hat{a}_\nu^\dagger \hat{a}_\nu$. The only two configurations satisfying $\sum_{\nu=1,2} \nu (\mathrm{mod}2) = 0$ are $(0,0)$ and $(1,1)$ corresponding to the two basis states $|1\rangle = \frac{1}{\sqrt{2}}(\hat{a}_0^\dagger)^2|0\rangle$ and $|2\rangle = \frac{1}{\sqrt{2}}(\hat{a}_1^\dagger)^2|0\rangle$, where $|0\rangle$ denotes the vacuum state. Since $(0,0)$ and $(1,1)$ are the only allowed configurations in the $P = 0$ sector, it follows from Ref. [10] that the larger value of $n_0$ and $n_1 = 2 - n_0$ is bounded from above by $2w_1$ and the lower one from below by $2w_2$. In particular, this means that the domain $\mathcal{E}_N^1(\boldsymbol{w})$ of the $\boldsymbol{w}$-ensemble functional $\mathcal{F}_{\boldsymbol{w}}$ is already convex,

$$\Sigma(\boldsymbol{w}, P = 0) = \{n_0 \,|\, 0 \leq 2w_2 \leq n_0 \leq 2w_1 \leq 2\}. \quad (9)$$

Then, minimizing the expectation value $\mathrm{Tr}_2[\hat{W}\hat{\Gamma}]$ according to the constrained search formalism, where $\hat{W}$ is the second term in the Hamiltonian (8), leads to (see Appendix A)

$$\mathcal{F}_{\boldsymbol{w}}(n_0) = U\left(1 - \sqrt{n_0(2 - n_0) - 4w_1 w_2}\right)$$
$$= U\left(1 - \sqrt{(n_0 - 2w_2)(2w_1 - n_0)}\right). \quad (10)$$

Since $\mathcal{F}_{\boldsymbol{w}}(n_0)$ is already convex, it is equal to the relaxed functional $\overline{\mathcal{F}}_{\boldsymbol{w}}(n_0)$.

The two equivalent expressions of $\mathcal{F}_{\boldsymbol{w}}(n_0)$ in Eq. (10) illustrate two different properties of the universal functional. From the first line together with (9) it follows immediately that $\mathcal{F}_{\boldsymbol{w}}(n_0)$ is symmetric around $n_0 = 1$. The second expression in (10) emphasizes the diverging behaviour of the gradient of $\mathcal{F}_{\boldsymbol{w}}(n_0)$ at the boundary of its domain for each weight $w_1 = 1 - w_2$. Indeed, the derivative of $\mathcal{F}_{\boldsymbol{w}}(n_0)$ with respect to $n_0$ diverges at the boundary $\partial\Sigma$ of the domain of $\mathcal{F}_{\boldsymbol{w}}$ as

$$\left|\frac{\partial \mathcal{F}_{\boldsymbol{w}}(n_0)}{\partial n_0}\right| \sim \frac{1}{\sqrt{\mathrm{dist}(n_0, \partial\Sigma)}}. \quad (11)$$

The sign of the gradient reveals that the corresponding force is repulsive, i.e., it prevents $n_0$ from ever reaching the boundary $\partial\Sigma$.

The universal functional $\mathcal{F}_{\boldsymbol{w}}(n_0)$ is illustrated for several values of $w_1$ in Fig. 1. This also demonstrates the inclusion relation [9, 10]

$$\boldsymbol{w}' \prec \boldsymbol{w} \;\Leftrightarrow\; \bar{\mathcal{E}}_N^1(\boldsymbol{w}') \subset \overline{\mathcal{E}}_N^1(\boldsymbol{w}). \quad (12)$$

Indeed, for $\boldsymbol{w}' \prec \boldsymbol{w}$ (corresponding here to $w_1' \leq w_1$) we have $\Sigma(\boldsymbol{w}', P = 0) \subset \Sigma(\boldsymbol{w}, P = 0)$.

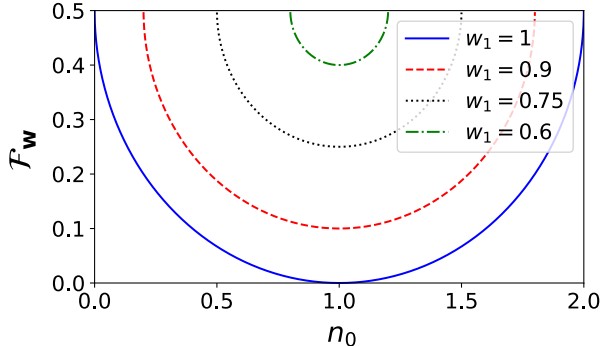

FIG. 1. Illustration of the $n_0$ dependence of the $\boldsymbol{w}$-ensemble functional $\mathcal{F}_{\boldsymbol{w}}$ for the symmetric Bose-Hubbard dimer with total momentum $P = 0$ for $U = 1/2$ and different values of the weight $w_1$ (recall that $w_1 + w_2 = 1$).

To further illustrate $\boldsymbol{w}$-ensemble RDMFT we calculate the energy $E_{\boldsymbol{w}}$ by minimizing the total energy functional $\mathrm{Tr}_1[\hat{t}\hat{\gamma}] + \mathcal{F}_{\boldsymbol{w}}(n_0)$, where $\hat{t}$ is given by the first term in (8) and $\mathrm{Tr}_1[\hat{t}\hat{\gamma}] = -2t(n_0 - 1)$. Solving

$$\left.\frac{\partial}{\partial n_0}\left(-2t(n_0 - 1) + \mathcal{F}_{\boldsymbol{w}}(n_0)\right)\right|_{n_0 = \tilde{n}_0} = 0 \quad (13)$$

for the minimizer $\tilde{n}_0$ and substituting the result into the energy functional yields for the weighted sum $E_{\boldsymbol{w}}$ of the two eigenenergies $E_1, E_2$ according to Eqs. (3) and (4),

$$E_{\boldsymbol{w}} = w_1\left(U - \sqrt{4t^2 + U^2}\right) + w_2\left(U + \sqrt{4t^2 + U^2}\right). \quad (14)$$

Note that this result is in agreement with the eigenenergies $E_1$ and $E_2$ obtained from an exact diagonalization of the Hamiltonian $\hat{H}$ in Eq. (8) (see Appendix A for further details).

## IV. $\boldsymbol{w}$-RDMFT FOR BOSE-EINSTEIN CONDENSATES

The application of $\boldsymbol{w}$-ensemble RDMFT to Bose-Einstein condensates (BECs) is appealing due to a number of reasons. First, as already stressed in the introduction, the Penrose and Onsager criterion [1, 3] for BEC identifies RDMFT as a particularly suitable approach to BECs. Second, recent analyses of their ground states have revealed an intriguing new concept, namely the existence of a BEC-force [3, 4, 6]. The question arises

whether this force based on the one-particle picture is also present in excited BECs. Third, the comprehensive understanding of the regime of small quantum depletion through Bogoliubov theory provides excellent prospects for deriving a corresponding approximation of the universal functional. This actually allows us to provide three conceptually different derivations of the $\boldsymbol{w}$-ensemble universal functional in the following. On an equal footing these three approaches illustrate how $\boldsymbol{w}$-ensemble universal functionals could be developed for fermions. For instance, recent advances in numerical techniques could be exploited to construct a universal functional via the Legendre-Fenchel transform.

To commence, we consider a dilute homogeneous BEC in a three-dimensional box with volume $V$. In second quantization, the general Hamiltonian $\hat{H} = \hat{h} + \hat{W}$ for interacting bosons in momentum representation reads

$$\hat{H} = \sum_{\boldsymbol{p}} t_{\boldsymbol{p}} \hat{a}_{\boldsymbol{p}}^{\dagger} \hat{a}_{\boldsymbol{p}} + \frac{1}{2V} \sum_{\boldsymbol{p},\boldsymbol{q},\boldsymbol{k}} W_{\boldsymbol{p}} \hat{a}_{\boldsymbol{q}+\boldsymbol{p}}^{\dagger} \hat{a}_{\boldsymbol{k}-\boldsymbol{p}}^{\dagger} \hat{a}_{\boldsymbol{k}} \hat{a}_{\boldsymbol{q}}, \quad (15)$$

where $\hat{a}_{\boldsymbol{p}}^{\dagger}$ and $\hat{a}_{\boldsymbol{p}}$ are the bosonic creation and annihilation operators, $W_{\boldsymbol{p}}$ the Fourier coefficients of the pair interaction between the bosons and $t_{\boldsymbol{p}}$ denotes the Fourier coefficients of the kinetic energy. By assuming a macroscopic occupation of the $\boldsymbol{p} = \boldsymbol{0}$ momentum state in the homogeneous BEC under consideration, the well-known Bogoliubov approximation [51] simplifies the general Hamiltonian in (15) to [52]

$$\hat{H}_{\mathrm{B}} = \sum_{\boldsymbol{p}} t_{\boldsymbol{p}} \hat{n}_{\boldsymbol{p}} + \frac{1}{2V} \sum_{\boldsymbol{p} \neq \boldsymbol{0}} W_{\boldsymbol{p}} \left[ 2\hat{n}_{\boldsymbol{0}} \hat{n}_{\boldsymbol{p}} + \left( \hat{a}_{\boldsymbol{p}}^{\dagger} \hat{a}_{-\boldsymbol{p}}^{\dagger} \hat{a}_{\boldsymbol{0}}^2 + \text{h.c.} \right) \right],$$
$$(16)$$

where $\hat{n}_{\boldsymbol{p}} = \hat{a}_{\boldsymbol{p}}^{\dagger} \hat{a}_{\boldsymbol{p}}$ is the occupation number operator and we omit the constant energy shift $\frac{N(N-1)W_{\boldsymbol{0}}}{2V}$. Moreover, the Fourier coefficients satisfy $W_{\boldsymbol{p}} = W_{-\boldsymbol{p}}$ and we restrict to purely repulsive interactions such that $W_{\boldsymbol{p}} \geq 0 \ \forall \boldsymbol{p}$. Furthermore, $t_{\boldsymbol{p}} = t_{-\boldsymbol{p}}$ with $t_{\boldsymbol{0}} = 0$. This implies in particular that the Bogoliubov approximated Hamiltonian (16) is invariant under $\boldsymbol{p} \to -\boldsymbol{p}$. We therefore introduce a new index $\boldsymbol{p}'$ which labels all pairs $(-\boldsymbol{p}, \boldsymbol{p}), \boldsymbol{p} \neq \boldsymbol{0}$ and for each such pair $\boldsymbol{p}'$ can be chosen to be either $\boldsymbol{p}' = -\boldsymbol{p}$ or $\boldsymbol{p}' = \boldsymbol{p}$ without loss of generality. Then, the Hamiltonian $\hat{H}_{\mathrm{B}}$ in Eq. (16) is equivalent to

$$\hat{H}_{\mathrm{B}} = \sum_{\boldsymbol{p}'} t_{\boldsymbol{p}'} \hat{\eta}_{\boldsymbol{p}'} + \frac{1}{V} \sum_{\boldsymbol{p}'} W_{\boldsymbol{p}'} \left[ \hat{n}_{\boldsymbol{0}} \hat{\eta}_{\boldsymbol{p}'} + \left( \hat{a}_{\boldsymbol{p}'}^{\dagger} \hat{a}_{-\boldsymbol{p}'}^{\dagger} \hat{a}_{\boldsymbol{0}}^2 + \text{h.c.} \right) \right],$$
$$(17)$$

where we introduced the operator

$$\hat{\eta}_{\boldsymbol{p}'} \equiv \hat{n}_{\boldsymbol{p}'} + \hat{n}_{-\boldsymbol{p}'}. \quad (18)$$

This notation emphasises that the expectation value of the kinetic energy operator

$$\mathrm{Tr}_1[\hat{t}\hat{\gamma}] = \sum_{\boldsymbol{p}} t_{\boldsymbol{p}} n_{\boldsymbol{p}}$$
$$= \sum_{\boldsymbol{p}'} t_{\boldsymbol{p}'} (n_{\boldsymbol{p}'} + n_{-\boldsymbol{p}'}) \equiv \sum_{\boldsymbol{p}'} t_{\boldsymbol{p}'} \eta_{\boldsymbol{p}'} \quad (19)$$

is completely determined by the pairs $(t_{\boldsymbol{p}'}, \eta_{\boldsymbol{p}'})$ for all $\boldsymbol{p}'$ since $t_{\boldsymbol{p}} = t_{-\boldsymbol{p}}$. In particular, this implies that the vectors $\boldsymbol{t} \equiv (t_{\boldsymbol{p}'})_{\boldsymbol{p}'}$ and $\boldsymbol{\eta} \equiv (\eta_{\boldsymbol{p}'})_{\boldsymbol{p}'}$ constitute the conjugate variables in our functional theory, denoted by $\boldsymbol{t} \leftrightarrow \boldsymbol{\eta}$. Furthermore, it follows that the ground state functional $\mathcal{F}_{\boldsymbol{w}_0}$ (recall that $\boldsymbol{w}_0 = (1, 0, \dots)$) and the excited state functional $\mathcal{F}_{\boldsymbol{w}}$ can be written as functionals of $\boldsymbol{\eta}$ only, i.e. $\mathcal{F}_{\boldsymbol{w}_0} \equiv \mathcal{F}_{\boldsymbol{w}_0}(\boldsymbol{\eta})$ and $\mathcal{F}_{\boldsymbol{w}} \equiv \mathcal{F}_{\boldsymbol{w}}(\boldsymbol{\eta})$.

## A. Recap of ground state universal functional

In this section, we derive the ground state universal functional $\mathcal{F}_{\boldsymbol{w}_0}$. The calculation shown in the following uses the same concepts as in Ref. [3] but derives $\mathcal{F}_{\boldsymbol{w}_0}$ as a functional of $\boldsymbol{\eta}$ rather than of the full occupation number vector $\boldsymbol{n}$.

In the Bogoliubov theory, the interacting ground state of a BEC has the form $|\Psi_0\rangle = \hat{U}|N\rangle$, where $|N\rangle = (N!)^{-1/2} (\hat{a}_{\boldsymbol{0}}^{\dagger})^N |0\rangle$ and [52, 53]

$$\hat{U} = \exp\left\{ \frac{1}{2} \sum_{\boldsymbol{p} \neq \boldsymbol{0}} \theta_{\boldsymbol{p}} \left[ (\hat{\beta}_{\boldsymbol{0}}^{\dagger})^2 \hat{a}_{\boldsymbol{p}} \hat{a}_{-\boldsymbol{p}} - \hat{\beta}_{\boldsymbol{0}}^2 \hat{a}_{\boldsymbol{p}}^{\dagger} \hat{a}_{-\boldsymbol{p}}^{\dagger} \right] \right\} \quad (20)$$

is a unitary operator with variational parameters $\theta_{\boldsymbol{p}} \in \mathbb{R}$. Here, the operator $\hat{\beta}_{\boldsymbol{0}} \equiv (\hat{n}_{\boldsymbol{0}} + 1)^{-1/2} \hat{a}_{\boldsymbol{0}}$ [53] annihilates a boson with momentum $\boldsymbol{p} = \boldsymbol{0}$ without creating a prefactor in front of the new state. In particular, $\hat{U}$ commutes with the particle number operator, $[\hat{U}, \hat{N}] = 0$. Moreover, the operators $\hat{\beta}_{\boldsymbol{0}}, \hat{\beta}_{\boldsymbol{0}}^{\dagger}$ ensure that the Hamiltonian $\hat{H}_{\mathrm{B}}$ in Eq. (16) still commutes with the particle number operator $\hat{N}$ after the Bogoliubov transformation and that the interacting ground state, i.e. the Bogoliubov quasi-particle vacuum, is a state in the $N$-boson Hilbert space. Then, the $\boldsymbol{w}$-minimizer for $r = 1$ (referring to ground state RDMFT) is given by $\hat{\Gamma}_{\boldsymbol{w}_0} = |\Psi_0\rangle\langle\Psi_0|$ according to the GOK variational principle in Eq. (3). Usually it is assumed that $\theta_{\boldsymbol{p}} = \theta_{-\boldsymbol{p}}$. If we allowed, however, for $\theta_{\boldsymbol{p}} \neq \theta_{-\boldsymbol{p}}$ one could show that

$$\hat{U}^{\dagger} \hat{a}_{\boldsymbol{p}} \hat{U} \approx \frac{1}{\sqrt{1 - \phi_{\boldsymbol{p}}^2}} \left( \hat{a}_{\boldsymbol{p}} - \phi_{\boldsymbol{p}} \hat{\beta}_{\boldsymbol{0}}^2 \hat{a}_{-\boldsymbol{p}}^{\dagger} \right) \quad (21)$$

with variational parameters

$$\phi_{\boldsymbol{p}} = \tanh\left( \frac{\theta_{\boldsymbol{p}} + \theta_{-\boldsymbol{p}}}{2} \right) \quad (22)$$

satisfying also $\phi_{\boldsymbol{p}} = \phi_{-\boldsymbol{p}}$. To proceed, the ground state functional $\mathcal{F}_{\boldsymbol{w}_0}$ is obtained by minimizing the expectation value $\langle\Psi_0|\hat{W}_{\mathrm{B}}|\Psi_0\rangle = \langle N|\hat{U}^{\dagger}\hat{W}_{\mathrm{B}}\hat{U}|N\rangle$ over the variational parameters $\phi_{\boldsymbol{p}'}$. Here, $\hat{W}_{\mathrm{B}}$ denotes the Bogoliubov approximated interaction $\hat{W}_{\mathrm{B}} = \hat{H}_{\mathrm{B}} - \hat{t}$, where $\hat{t} = \sum_{\boldsymbol{p}'} t_{\boldsymbol{p}'} \hat{\eta}_{\boldsymbol{p}'}$ is the kinetic energy operator. It is a straightforward calculation to express the expectation value $\langle\Psi_0|\hat{W}_{\mathrm{B}}|\Psi_0\rangle$ in terms of $\{\phi_{\boldsymbol{p}'}\}_{\boldsymbol{p}'}$ by inserting identities $\hat{\mathbb{1}} = \hat{U}^{\dagger}\hat{U}$ and using Eq. (21) as demonstrated in

Ref. [52]. From Eq. (22) and the definition of the ground state $|\Psi_0\rangle$ it follows that $\eta_{\boldsymbol{p}'}$ and $\phi_{\boldsymbol{p}'}$ are related through

$$\eta_{\boldsymbol{p}'} \equiv \langle\Psi_0|\hat{\eta}_{\boldsymbol{p}'}|\Psi_0\rangle \approx \frac{2\phi_{\boldsymbol{p}'}^2}{1 - \phi_{\boldsymbol{p}'}^2} \,. \tag{23}$$

The expression on the right hand side holds only approximately due to the approximation in Eq. (21) and an estimate of its accuracy can be deduced from Ref. [54]. Inverting this expression for the variational parameters $\phi_{\boldsymbol{p}'}$ shows that the occupation numbers $\eta_{\boldsymbol{p}'}$ determine the variational parameters $\phi_{\boldsymbol{p}'}$ up to phases $\sigma_{\boldsymbol{p}'} = \pm 1$. This simplifies the minimization in Levy's constrained search (4) over all $\phi_{\boldsymbol{p}'}$ to a minimization over the phases $\sigma_{\boldsymbol{p}'}$ according to

$$\mathcal{F}_{\boldsymbol{w}_0}(\boldsymbol{\eta}) = \min_{\{\sigma_{\boldsymbol{p}'} = \pm 1\}} \left[ n\sum_{\boldsymbol{p}'} W_{\boldsymbol{p}'}\eta_{\boldsymbol{p}'} - \sigma_{\boldsymbol{p}'}W_{\boldsymbol{p}'}\sqrt{\eta_{\boldsymbol{p}'}(\eta_{\boldsymbol{p}'}+2)} \right]$$
$$= n\sum_{\boldsymbol{p}'} W_{\boldsymbol{p}'}\left( \eta_{\boldsymbol{p}'} - \sqrt{\eta_{\boldsymbol{p}'}(\eta_{\boldsymbol{p}'}+2)} \right) , \tag{24}$$

where we used $W_{\boldsymbol{p}'} \geq 0 \; \forall \boldsymbol{p}'$ in the last line and $n = N/V$ denotes the particle density. As a consistency check, we note that the universal functional (24) is indeed equivalent to the one derived in Ref. [3] after replacing in the latter the momentum occupation numbers $n_{\boldsymbol{p}}$ by $\eta_{\boldsymbol{p}'}/2$.

## B. Excitations within Bogoliubov theory

In the following we recall the most important aspects of the excitation spectrum of a homogeneous Bose gas within the Bogoliubov approximation. This serves as a preliminary for the derivation of the excited state functional $\mathcal{F}_{\boldsymbol{w}}$ in Sec. IV E.

The ground state energy of the Bogoliubov approximated Hamiltonian $\hat{H}_\mathrm{B}$ is given by [52, 54]

$$E_0 = -\frac{1}{2}\sum_{\boldsymbol{p}\neq\boldsymbol{0}} \left( nW_{\boldsymbol{p}} + t_{\boldsymbol{p}} - \sqrt{t_{\boldsymbol{p}}(t_{\boldsymbol{p}}+2nW_{\boldsymbol{p}})} \right) , \tag{25}$$

and the same result also holds approximately within the particle number conserving Bogoliubov theory up to a controllable error [52, 54]. Moreover, the energy spectrum consists of elementary excitations of the ground state and takes the form [51, 52]

$$E = E_0 + \sum_{\boldsymbol{p}\neq\boldsymbol{0}} \omega_{\boldsymbol{p}}\mu_{\boldsymbol{p}} \,. \tag{26}$$

Here, $\mu_{\boldsymbol{p}}$ counts the number of quasiparticles with momentum $\boldsymbol{p}$ created by acting with the quasiparticle operator [52]

$$\hat{c}_{\boldsymbol{p}}^\dagger \equiv \hat{U}\hat{a}_{\boldsymbol{p}}^\dagger\hat{U}^\dagger\hat{\beta}_0$$
$$\approx \frac{1}{\sqrt{1-\phi_{\boldsymbol{p}}^2}}\left( \hat{a}_{\boldsymbol{p}}^\dagger\hat{\beta}_0 + \phi_{\boldsymbol{p}}\hat{\beta}_0^\dagger\hat{a}_{-\boldsymbol{p}} \right) \tag{27}$$

on the interacting ground state $|\Psi_0\rangle$, and $\omega_{\boldsymbol{p}}$ denotes the quasiparticle dispersion relation

$$\omega_{\boldsymbol{p}} = \sqrt{t_{\boldsymbol{p}}(t_{\boldsymbol{p}}+2nW_{\boldsymbol{p}})} \,. \tag{28}$$

For small enough quantum depletion and low-lying excited states, (26) holds in good approximation also for the particle number conserving Bogoliubov Hamiltonian with (16) and, in particular, $\mu_{\boldsymbol{p}} \approx n_{\boldsymbol{p}}$ in Eq. (26) [54]. Furthermore, $\omega_{\boldsymbol{p}} = \omega_{-\boldsymbol{p}}$ and we replace accordingly $\boldsymbol{p}$ by $\boldsymbol{p}'$ in the derivation of the $\boldsymbol{w}$-ensemble functional.

Before we can present three different instructive derivations of the $\boldsymbol{w}$-ensemble functional $\mathcal{F}_{\boldsymbol{w}}$ for targeting the ground state and the first excited state, we discuss two critical conceptual aspects of $\boldsymbol{w}$-ensemble RDMFT. Both Secs. IV C and IV D and their conclusions are not restricted to BECs but are valid for $\boldsymbol{w}$-RDMFT applied to arbitrary quantum systems of bosons or fermions.

## C. Crossing of energy levels

The energy levels of many-body quantum systems can cross as one varies system parameters such as the coupling constants of two-body interactions or the strength of an external field. A particularly prominent example is given by quantum phase transitions for which the ground state and first excited state cross. This in turn manifests itself in the context of functional theories in the form of nonanalyticities of the universal functional: By referring to the constrained search formalism, the $N$-fermion minimizer for 1RDMs belonging to different quantum phases are not necessarily analytically connected anymore. As a consequence, the functional's domain would split into different cells (subdomains) and one would need to derive an analytical functional for each of them separately. At the borders of those cells those different functionals would be "glued" together continuously. This adds to several further consequences of level crossings discussed in functional theories [55–57].

In the context of excited state RDMFT this reasoning would apply to various energy levels of interest, i.e., the lowest $r$ ones in $\boldsymbol{w}$-RDMFT. Accordingly, there will be many more relevant crossings and the functional's domain would divide into even more cells than in case of ground state RDMFT. These consequences of crossing energy levels make the calculation of the universal functional in the following more involved. From a general perspective, this highlights that the commonly pursued strategy to write down smooth ansatzes for the universal functional is rather problematic. At the same time, it also questions the importance and meaning of universality in functional theories.

## D. $\boldsymbol{w}$-ensemble $v$-representability problem

The original formulation of ground state RDMFT by Gilbert [58] was hampered by the so-called $v$-

representability problem which for most quantum systems is impossible to solve. A 1RDM $\hat{\gamma} \in \mathcal{P}_N^1$ is called $v$-representable if there exists some one-particle Hamiltonian $\hat{h}$ yielding $\hat{\gamma}_{\hat{h}}$ as the ground state 1RDM according to

$$\hat{h} \mapsto \hat{H}(\hat{h}) \mapsto |\Psi_{\hat{h}}\rangle \mapsto \hat{\gamma}_{\hat{h}} \,, \tag{29}$$

where $|\Psi_{\hat{h}}\rangle$ denotes the $N$-particle ground state of $\hat{H}(\hat{h})$ (1). The significance of this definition rests upon the following relation between the ground state energy $E$ and the universal ground state functional $\mathcal{F}$ for $v$-representable 1RDMs,

$$\mathcal{F}(\hat{\gamma}_{\hat{h}}) = E(\hat{\gamma}_{\hat{h}}) - \text{Tr}_1[\hat{\gamma}_{\hat{h}} \hat{h}] \,. \tag{30}$$

Because of its fruitful consequence (30), we now establish an extension of $v$-representability to $\boldsymbol{w}$-ensemble RDMFT. A 1RDM $\hat{\gamma} \in \mathcal{E}_N^1(\boldsymbol{w})$ shall be called $\boldsymbol{w}$-ensemble $v$-representable if $\hat{\gamma}$ emerges as the 1RDM of the minimizer in the GOK variational principle (3) applied to the Hamiltonian $\hat{H}(\hat{h})$ (1) for some $\hat{h}$. Note that in Sec. II, this $\boldsymbol{w}$-ensemble $v$-representability problem was circumvented by the constrained search formalism (4) from the very beginning. Yet, if there was given a compact solution of the $\boldsymbol{w}$-ensemble $v$-representability problem, the universal functional could be determined more directly. In analogy to (30), for any $\boldsymbol{w}$-ensemble $v$-representable $\hat{\gamma}_{\hat{h}} \in \mathcal{E}_N^1(\boldsymbol{w})$, the $\boldsymbol{w}$-ensemble functional follows from the energy $E_{\boldsymbol{w}}$ (3) as

$$\mathcal{F}_{\boldsymbol{w}}(\hat{\gamma}_{\hat{h}}) = E_{\boldsymbol{w}}(\hat{h}) - \text{Tr}_1[\hat{\gamma}_{\hat{h}} \hat{h}] \,. \tag{31}$$

In Sec. IV E 2, we illustrate the derivation of $\mathcal{F}_{\boldsymbol{w}}$ for all $\boldsymbol{w}$-ensemble $v$-representable 1RDMs for a homogeneous BEC.

The two $\boldsymbol{w}$-ensemble functionals $\mathcal{F}_{\boldsymbol{w}}$ and $\overline{\mathcal{F}}_{\boldsymbol{w}}$ defined in Eqs. (4) and (6) are equal for a given 1RDM $\hat{\gamma}$ whenever $\hat{\gamma}$ is $\boldsymbol{w}$-ensemble $v$-representable. In case $\mathcal{F}_{\boldsymbol{w}}$ is convex, every $\hat{\gamma} \in \mathcal{E}_N^1(\boldsymbol{w})$[59] is $\boldsymbol{w}$-ensemble $v$-representable, a statement that is well known in the context of ground state functional theory (see, e.g., Ref. [20, 60]). Since the Legendre-Fenchel transform of $\mathcal{F}_{\boldsymbol{w}}$ is the energy $E_{\boldsymbol{w}}$ up to minus signs, this implies that for a *convex* functional $\mathcal{F}_{\boldsymbol{w}}$ the biconjugate [61] $\mathcal{F}_{\boldsymbol{w}}^{**} \equiv \text{conv}(\mathcal{F}_{\boldsymbol{w}}) \equiv \overline{\mathcal{F}}_{\boldsymbol{w}}$ is equal to the functional $\mathcal{F}_{\boldsymbol{w}}$ itself. Moreover, as an alternative to the constrained search formalism (4), we can then derive not only $\overline{\mathcal{F}}_{\boldsymbol{w}}$ but also $\mathcal{F}_{\boldsymbol{w}}$ through a Legendre-Fenchel transformation. Conversely, if $\mathcal{F}_{\boldsymbol{w}}$ turns out to be non-convex, the set $\mathcal{E}_N^1(\boldsymbol{w})$ has to contain 1RDMs $\hat{\gamma}$ which are *not* $\boldsymbol{w}$-ensemble $v$-representable. In that case, calculating the biconjugate $\mathcal{F}_{\boldsymbol{w}}^{**}$ will just yield the lower convex envelope of $\mathcal{F}_{\boldsymbol{w}}$.

We will exploit the Legendre-Fenchel transformation in Sec. IV E 1 to derive the $\boldsymbol{w}$-ensemble functional $\overline{\mathcal{F}}_{\boldsymbol{w}}$ for a homogeneous BEC.

## E. Derivation of $\boldsymbol{w}$-ensemble functional for $r = 2$

In order to apply the $\boldsymbol{w}$-ensemble RDMFT for bosons to a homogeneous BEC, we restrict in the following to finite but large enough systems such that the $\boldsymbol{p} = \boldsymbol{0}$ momentum state is macroscopically occupied and there exists a finite gap between the energy levels. Due to $W_{\boldsymbol{p}} = W_{-\boldsymbol{p}}$ and $t_{\boldsymbol{p}} = t_{-\boldsymbol{p}}$, the excited energy states are degenerate. In the following, we restrict to $r = 2$ non-vanishing weights $w_j$ such that the corresponding $\boldsymbol{w}$-ensemble functional $\mathcal{F}_{\boldsymbol{w}}$ in (4) allows one to determine the ground state and the first excited state. Thus, we consider weight vectors of the form

$$\boldsymbol{w} = (w, 1 - w, 0, \dots) \tag{32}$$

with $w \geq \frac{1}{2}$. According to Eq. (26), for each $\boldsymbol{p}'$ the weighted sum of the ground state energy $E_0$ (25) and a single excitation with momentum $\boldsymbol{p}'$ reads

$$E_{\boldsymbol{w}, \boldsymbol{p}'} = w E_0 + (1 - w) E_1 = E_0 + (1 - w)\omega_{\boldsymbol{p}'} \,. \tag{33}$$

This implies that the sought-after energy $E_{\boldsymbol{w}}$ follows as

$$E_{\boldsymbol{w}} = \min_{\boldsymbol{p}'} E_{\boldsymbol{w}, \boldsymbol{p}'} \,. \tag{34}$$

Furthermore, the domain of the relaxed $\boldsymbol{w}$-ensemble functional $\overline{\mathcal{F}}_{\boldsymbol{w}}$ is given by the spectral polytope [10]

$$\Sigma(\boldsymbol{w}) = \text{conv}\left( \{\pi(\boldsymbol{v}) \,|\, \pi \in \mathcal{S}^d\} \right) \,, \tag{35}$$

where $d$ denotes the dimension of the one-particle Hilbert space $\mathcal{H}_1$, $\mathcal{S}^d$ the permutation group of $d$ elements and $\boldsymbol{v}$ is the natural occupation number vector

$$\boldsymbol{v} = (N - 1 + w, 1 - w, 0, \dots) \,. \tag{36}$$

In the following, we present three different approaches for deriving the universal $\boldsymbol{w}$-ensemble functional for $r = 2$ non-vanishing weights in the context of Bogoliubov theory, i.e., in the regime of small quantum depletion. This also allows us to illustrate various aspects of $\boldsymbol{w}$-ensemble RDMFT discussed in the previous sections.

### 1. Legendre-Fenchel transformation

In the following, we derive the relaxed $\boldsymbol{w}$-ensemble functional $\overline{\mathcal{F}}_{\boldsymbol{w}}$ for $r = 2$ non-vanishing weights through a Legendre-Fenchel transformation of the energy in Eq. (33). As introduced in Ref. [20] for the ground state functional and anticipated in Sec. IV D, the energy $E_{\boldsymbol{w}}$ and the $\boldsymbol{w}$-ensemble functional $\overline{\mathcal{F}}_{\boldsymbol{w}}$ are related through the Legendre-Fenchel transform by

$$\overline{\mathcal{F}}_{\boldsymbol{w}}^*(\hat{h}) \equiv \max_{\hat{\gamma} \in \overline{\mathcal{E}}_N^1(\boldsymbol{w})} \left[ \text{Tr}_1[\hat{h}\hat{\gamma}] - \overline{\mathcal{F}}_{\boldsymbol{w}}(\hat{\gamma}) \right] = -E_{\boldsymbol{w}}(-\hat{h}) \,. \tag{37}$$

Consequently, the biconjugate $\overline{\mathcal{F}}_{\boldsymbol{w}}^{**} \equiv (\overline{\mathcal{F}}_{\boldsymbol{w}}^*)^* = \overline{\mathcal{F}}_{\boldsymbol{w}}$ [61] of the convex $\overline{\mathcal{F}}_{\boldsymbol{w}}$ can be expressed as

$$\overline{\mathcal{F}}_{\boldsymbol{w}}(\hat{\gamma}) = \max_{\hat{h}} \left[ E_{\boldsymbol{w}}(\hat{h}) - \text{Tr}_1[\hat{h}\hat{\gamma}] \right] \,. \tag{38}$$

Since the energy $E_{\boldsymbol{w}}$ is related to the energies $E_{\boldsymbol{w},\boldsymbol{p}'}$ (see Eq. (34)), auxiliary functionals $\overline{\mathcal{F}}_{\boldsymbol{w},\boldsymbol{p}'}$ are introduced as the Legendre-Fenchel transforms (up to the common minus signs) of the (concave) $E_{\boldsymbol{w},\boldsymbol{p}'}$ for all $\boldsymbol{p}'$. They allow us to rewrite the energy $E_{\boldsymbol{w}}$ using Eq. (34) as

$$E_{\boldsymbol{w}}(\hat{h}) = \min_{\boldsymbol{p}'} \min_{\hat{\gamma} \in \bar{\mathcal{E}}_N^1(\boldsymbol{w})} \left[ \mathrm{Tr}_1[\hat{h}\hat{\gamma}] + \overline{\mathcal{F}}_{\boldsymbol{w},\boldsymbol{p}'}(\hat{\gamma}) \right]$$

$$= \min_{\hat{\gamma} \in \bar{\mathcal{E}}_N^1(\boldsymbol{w})} \left[ \mathrm{Tr}_1[\hat{h}\hat{\gamma}] + \min_{\boldsymbol{p}'} \overline{\mathcal{F}}_{\boldsymbol{w},\boldsymbol{p}'}(\hat{\gamma}) \right]. \quad (39)$$

The expression $\min_{\boldsymbol{p}'} \overline{\mathcal{F}}_{\boldsymbol{w},\boldsymbol{p}'}$ can be interpreted as a universal functional and it coincides with $\overline{\overline{\mathcal{F}}}_{\boldsymbol{w}}$ up to a lower convex envelop,

$$\overline{\overline{\mathcal{F}}}_{\boldsymbol{w}} = \mathrm{conv}\left( \min_{\boldsymbol{p}'} \overline{\mathcal{F}}_{\boldsymbol{w},\boldsymbol{p}'} \right). \quad (40)$$

It is worth noticing here that the minimum of a family of convex functions (e.g., $\{\overline{\mathcal{F}}_{\boldsymbol{w},\boldsymbol{p}'}\}$) is not necessarily convex [62]. Moreover, the second line in Eq. (39)

To proceed, we first recall that we restrict to $\hat{h} \equiv \hat{t}$ with $t_{\boldsymbol{p}} = t_{-\boldsymbol{p}}$ which implies that the inner product $\langle \hat{\gamma}, \hat{t} \rangle$ for a fixed $\hat{t}$ is completely determined through the vector $\boldsymbol{\eta}$ defined in Eq. (18). Then, the maximum in Eq. (38) is obtained by solving for all $\tilde{\boldsymbol{p}}'$

$$\eta_{\tilde{\boldsymbol{p}}'} \equiv n_{\tilde{\boldsymbol{p}}} + n_{-\tilde{\boldsymbol{p}}} = \frac{\partial E_{\boldsymbol{w},\boldsymbol{p}'}(\hat{t})}{\partial t_{\tilde{\boldsymbol{p}}'}}. \quad (41)$$

Its solution $t_{\tilde{\boldsymbol{p}}'}(\eta_{\tilde{\boldsymbol{p}}'})$ corresponding to a maximum reads

$$t_{\tilde{\boldsymbol{p}}'}(\eta_{\tilde{\boldsymbol{p}}'}) = \begin{cases} nW_{\tilde{\boldsymbol{p}}'}\left( \frac{1+\eta_{\tilde{\boldsymbol{p}}'}}{\sqrt{\eta_{\tilde{\boldsymbol{p}}'}(\eta_{\tilde{\boldsymbol{p}}'}+2)}} - 1 \right) & \text{if } \tilde{\boldsymbol{p}}' \neq \boldsymbol{p}', \\ nW_{\tilde{\boldsymbol{p}}'}\left( \frac{1+\eta_{\tilde{\boldsymbol{p}}'}}{\sqrt{(\eta_{\tilde{\boldsymbol{p}}'}+3-w)(\eta_{\tilde{\boldsymbol{p}}'}+w-1)}} - 1 \right) & \text{if } \tilde{\boldsymbol{p}}' = \boldsymbol{p}'. \end{cases}$$

$$(42)$$

At this point, we would like to recall that the momenta $\boldsymbol{p}'$ label the pairs $(-\boldsymbol{p},\boldsymbol{p}), \boldsymbol{p} \neq \boldsymbol{0}$ as defined above Eq. (16). Combining (38) and (42) eventually leads to

$$\overline{\mathcal{F}}_{\boldsymbol{w},\boldsymbol{p}'}(\boldsymbol{\eta}) = \mathcal{F}_{\boldsymbol{w}_0}(\boldsymbol{\eta}) + nW_{\boldsymbol{p}'}\left( \sqrt{\eta_{\boldsymbol{p}'}(\eta_{\boldsymbol{p}'}+2)} - \sqrt{(\eta_{\boldsymbol{p}'}+3-w)(\eta_{\boldsymbol{p}'}+w-1)} \right). \quad (43)$$

For a given $\boldsymbol{p}'$ the functional $\overline{\mathcal{F}}_{\boldsymbol{w},\boldsymbol{p}'}(\boldsymbol{\eta})$ equals the universal functional $\overline{\overline{\mathcal{F}}}_{\boldsymbol{w}}(\boldsymbol{\eta})$ only for those $\boldsymbol{\eta}$ whose minimizers in (6) involve as first excited state the respective $\boldsymbol{p}'$-excitation. In general, the functional $\overline{\overline{\mathcal{F}}}_{\boldsymbol{w}}$ then follows from Eq. (40). In particular, based on the form (43), one can verify that $\min_{\boldsymbol{p}'} \overline{\mathcal{F}}_{\boldsymbol{w},\boldsymbol{p}'}$ is typically not convex and thus the lower convex envelop operation $\mathrm{conv}(\cdot)$ in Eq. (40) is essential.

We close this section by observing the following intriguing relation. $\overline{\overline{\mathcal{F}}}_{\boldsymbol{w}}$ (through $\overline{\mathcal{F}}_{\boldsymbol{w},\boldsymbol{p}'}$) consists of the convex ground state functional $\overline{\overline{\mathcal{F}}}_{\boldsymbol{w}_0}$ given by Eq. (24) plus an additional positive term which always increases the energy due to a single elementary excitation $\boldsymbol{p}' \equiv \boldsymbol{p}'(\boldsymbol{\eta})$ of the ground state. Moreover, one can easily check that $\overline{\mathcal{F}}_{\boldsymbol{w},\boldsymbol{p}'}$ in Eq. (43) and thus $\overline{\overline{\mathcal{F}}}_{\boldsymbol{w}}$ in Eq. (40) reduces to $\overline{\overline{\mathcal{F}}}_{\boldsymbol{w}_0}$ for $w = 1$, as required.

### 2. Derivation of $\mathcal{F}_{\boldsymbol{w}}$ and $\overline{\mathcal{F}}_{\boldsymbol{w}}$ for all $\boldsymbol{w}$-ensemble v-representable 1RDMs

Once the energy $E_{\boldsymbol{w}}$ is known, we can derive for all $\boldsymbol{w}$-ensemble $v$-representable 1RDMs $\hat{\gamma}$ the value of the $\boldsymbol{w}$-ensemble functionals $\mathcal{F}_{\boldsymbol{w}}(\hat{\gamma}) = \overline{\mathcal{F}}_{\boldsymbol{w}}(\hat{\gamma})$ through Eq. (31)

In order to apply this approach to the Bogoliubov approximated interaction $\hat{W}_{\mathrm{B}}$, we first define the momentum $\boldsymbol{q}'$ corresponding to the first excitation, which is determined through the lowest value of the quasiparticle

and Eq. 40 reflect very well the curse of universality outlined in Sec. IV C: no closed analytical form exists for $\min_{\boldsymbol{p}'} \overline{\mathcal{F}}_{\boldsymbol{w},\boldsymbol{p}'}$ and $\overline{\overline{\mathcal{F}}}_{\boldsymbol{w}}$, respectively.

dispersion $\omega_{\boldsymbol{p}} = \omega_{-\boldsymbol{p}}$. Moreover, the degenerate subspace of the first excited state $|\Psi_1\rangle$ is spanned by the two orthonormal states $\hat{c}_{\boldsymbol{q}}^\dagger|\Psi_0\rangle$ and $\hat{c}_{-\boldsymbol{q}}^\dagger|\Psi_0\rangle$. Thus, any superposition state

$$|\Psi_1\rangle = \alpha \hat{c}_{\boldsymbol{q}}^\dagger|\Psi_0\rangle + \beta \hat{c}_{-\boldsymbol{q}}^\dagger|\Psi_0\rangle \quad (44)$$

with $\alpha, \beta \in \mathbb{C}$ and normalization $|\alpha|^2 + |\beta|^2 = 1$ corresponds to a single excitation on top of the interacting ground state $|\Psi_0\rangle$. Therefore, within the Bogoliubov approximation, we restrict the set $\mathcal{E}^N$ of all $N$-boson density operators $\hat{\Gamma}$ to the subset of all variational states of the form

$$\hat{\Gamma}_{\boldsymbol{w}} = w|\Psi_0\rangle\langle\Psi_0| + (1-w)|\Psi_1\rangle\langle\Psi_1| \quad (45)$$

with $|\Psi_1\rangle$ given by Eq. (44). This variational ansatz reduces the minimization on the right hand side of the GOK variational principle (3) applied to the Bogoliubov Hamiltonian $\hat{H}_{\mathrm{B}}$ to a minimization of the energy

$$\mathrm{Tr}_N[\hat{H}_{\mathrm{B}}\hat{\Gamma}_{\boldsymbol{w}}] \quad (46)$$

$$= \sum_{\boldsymbol{p}'} 2 \left( (nW_{\boldsymbol{p}'} + t_{\boldsymbol{p}'}) \frac{\phi_{\boldsymbol{p}'}^2}{1 - \phi_{\boldsymbol{p}'}^2} - nW_{\boldsymbol{p}'} \frac{\phi_{\boldsymbol{p}'}}{1 - \phi_{\boldsymbol{p}'}^2} \right)$$

$$+ (1-w) \left( (nW_{\boldsymbol{q}'} + t_{\boldsymbol{q}'}) \frac{1 + \phi_{\boldsymbol{q}'}^2}{1 - \phi_{\boldsymbol{q}'}^2} - nW_{\boldsymbol{q}'} \frac{2\phi_{\boldsymbol{q}'}}{1 - \phi_{\boldsymbol{q}'}^2} \right)$$

over the variational parameters $\{\phi_{\boldsymbol{p}'}\}_{\boldsymbol{p}'}$ defined in Sec. IV A. Eq. (46) was derived in an analogous man-

ner as expressing $\langle N|\hat{U}^\dagger \hat{W}_B \hat{U}|N\rangle$ in terms of $\{\phi_{\boldsymbol{p}'}\}_{\boldsymbol{p}'}$ in Sec. IV A. In particular, it can be shown by a straightforward calculation that $\text{Tr}_N[\hat{H}_B \hat{\Gamma}_{\boldsymbol{w}}]$ reduces to $\langle N|\hat{U}^\dagger \hat{W}_B \hat{U}|N\rangle$ for $w = 1$, i.e. in the case of ground state RDMFT. Performing the minimization of (46) for all momenta $\boldsymbol{p}'$ separately leads to the solution

$$\tilde{\phi}_{\boldsymbol{p}'} \equiv \frac{1}{n W_{\boldsymbol{p}'}} \left( t_{\boldsymbol{p}'} + n W_{\boldsymbol{p}'} - \sqrt{t_{\boldsymbol{p}'}(t_{\boldsymbol{p}'} + 2n W_{\boldsymbol{p}'})} \right) , \quad (47)$$

in agreement with Ref. [52, 53]. As a consistency check one can show that Eqs. (46) and (47) indeed lead to $E_{\boldsymbol{w}}$ in (33). Furthermore, the expectation value of the operator $\hat{\eta}_{\boldsymbol{p}'}$ (18) is given by

$$\eta_{\boldsymbol{p}'} = \text{Tr}_N[\hat{\eta}_{\boldsymbol{p}'} \hat{\Gamma}_{\boldsymbol{w}}]$$
$$= \begin{cases} \frac{2\phi_{\boldsymbol{p}'}^2}{1-\phi_{\boldsymbol{p}'}^2} & \text{if } \boldsymbol{p}' \neq \boldsymbol{q}' , \\ w\frac{2\phi_{\boldsymbol{p}'}^2}{1-\phi_{\boldsymbol{p}'}^2} + (1-w)\frac{1+3\phi_{\boldsymbol{p}'}^2}{1-\phi_{\boldsymbol{p}'}^2} & \text{if } \boldsymbol{p}' = \boldsymbol{q}' . \end{cases} \quad (48)$$

Inserting the minimizers $\tilde{\phi}_{\boldsymbol{p}'}$ (47) into (48) leads for all momenta $\boldsymbol{p}'$ to the same solution for the dispersion $t_{\boldsymbol{p}'}$ as in (42). This allows us to use (31),

$$\mathcal{F}_{\boldsymbol{w},\boldsymbol{q}'}(\boldsymbol{\eta}) = E_{\boldsymbol{w},\boldsymbol{q}'}(\boldsymbol{t}) - \sum_{\boldsymbol{p}'} t_{\boldsymbol{p}'} \eta_{\boldsymbol{p}'} \quad (49)$$

to calculate the $\boldsymbol{w}$-ensemble functional $\mathcal{F}_{\boldsymbol{w},\boldsymbol{q}'}$ for all $\boldsymbol{w}$-ensemble $v$-representable $\boldsymbol{\eta}$. Evaluating (49) eventually leads to the same expression for $\mathcal{F}_{\boldsymbol{w},\boldsymbol{q}'}$ as Eq. (43) derived in Sec. IV E 1.

Since $\mathcal{F}_{\boldsymbol{w}}$ is given on its domain of $v$-representable $\boldsymbol{\eta}$ by $\min_{\boldsymbol{p}'} \overline{\mathcal{F}}_{\boldsymbol{w},\boldsymbol{p}'}$ and since the latter is typically not convex on $\overline{\mathcal{E}}_N^1(\boldsymbol{w})$ (recall (5)), it follows (see, e.g., [60] and

Sec. IV D) that some $\boldsymbol{w}$-ensemble $N$-representable $\boldsymbol{\eta}$ are not $\boldsymbol{w}$-ensemble $v$-representable.

### 3. Constrained search formalism

The complexity of the domain of $v$-representable 1RDMs can be circumvented through the constrained search formalism (4). As it has been outlined in Sec. II, the latter namely establishes a universal functional $\mathcal{F}_{\boldsymbol{w}}$ on the larger domain $\mathcal{E}_N^1(\boldsymbol{w})$, or equivalently $\overline{\mathcal{F}}_{\boldsymbol{w}} \equiv \text{conv}(\mathcal{F}_{\boldsymbol{w}})$ on $\overline{\mathcal{E}}_N^1(\boldsymbol{w}) \equiv \text{conv}(\mathcal{E}_N^1(\boldsymbol{w}))$. Therefore, the approach to derive $\mathcal{F}_{\boldsymbol{w}}$ by exploiting the notion of $\boldsymbol{w}$-ensemble $v$-representability in Sec. IV E 2 and the constrained search formalism discussed below are quite different from a conceptual point of view.

To illustrate the constrained search (4) for a homogeneous BEC, we first need to calculate the expectation value $\text{Tr}_N[\hat{W}_B \hat{\Gamma}_{\boldsymbol{w}}] = \text{Tr}_N[\hat{H}_B \hat{\Gamma}_{\boldsymbol{w}}] - \text{Tr}_N[\hat{t} \hat{\Gamma}_{\boldsymbol{w}}]$. Since $\text{Tr}_N[\hat{H}_B \hat{\Gamma}_{\boldsymbol{w}}]$ is given by Eq. (46) we immediately arrive at

$$\text{Tr}_N[\hat{W}_B \hat{\Gamma}_{\boldsymbol{w}}] = \sum_{\boldsymbol{p}'} 2n W_{\boldsymbol{p}'} \left( \frac{\phi_{\boldsymbol{p}'}^2}{1-\phi_{\boldsymbol{p}'}^2} - \frac{\phi_{\boldsymbol{p}'}}{1-\phi_{\boldsymbol{p}'}^2} \right) \quad (50)$$
$$+ (1-w)n W_{\boldsymbol{q}'} \left( \frac{1+\phi_{\boldsymbol{q}'}^2}{1-\phi_{\boldsymbol{q}'}^2} - \frac{2\phi_{\boldsymbol{q}'}}{1-\phi_{\boldsymbol{q}'}^2} \right) .$$

Furthermore, the occupation numbers $\eta_{\boldsymbol{p}'}$ in Eq. (48) determine the variational parameters $\phi_{\boldsymbol{p}'}$,

$$\phi_{\boldsymbol{p}'} = \begin{cases} \sigma_{\boldsymbol{p}'} \sqrt{\frac{\eta_{\boldsymbol{p}'}}{2+\eta_{\boldsymbol{p}'}}} & \text{if } \boldsymbol{p}' \neq \boldsymbol{q}' , \\ \sigma_{\boldsymbol{p}'} \sqrt{\frac{\eta_{\boldsymbol{p}'}+w-1}{\eta_{\boldsymbol{p}'}+3-w}} & \text{if } \boldsymbol{p}' = \boldsymbol{q}' , \end{cases} \quad (51)$$

up to a phase $\sigma_{\boldsymbol{p}'} = \pm 1$. As a result, the constrained search formalism (4) simplifies to a minimization over the phases $\sigma_{\boldsymbol{p}'}$ of $\phi_{\boldsymbol{p}'}$,

$$\mathcal{F}_{\boldsymbol{w},\boldsymbol{q}'}(\boldsymbol{\eta}) = \min_{\{\sigma_{\boldsymbol{p}'}=\pm 1\}} \left[ \sum_{\boldsymbol{p}'\neq\boldsymbol{q}'} n W_{\boldsymbol{p}'} \left( \eta_{\boldsymbol{p}'} - \sigma_{\boldsymbol{p}'} \sqrt{\eta_{\boldsymbol{p}'}(\eta_{\boldsymbol{p}'}+2)} \right) + n W_{\boldsymbol{q}'} \left( \eta_{\boldsymbol{q}'} - \sigma_{\boldsymbol{q}'} \sqrt{(\eta_{\boldsymbol{q}'}+3-w)(\eta_{\boldsymbol{q}'}+w-1)} \right) \right] , \quad (52)$$

which can be solved independently of the sign of the Fourier coefficients $W_{\boldsymbol{p}'}$. Here, we have $W_{\boldsymbol{p}'} \geq 0$ for all momenta $\boldsymbol{p}'$ such that the minimization in (52) leads indeed to the functional presented in Eq. (43). It is worth noticing that the minimization over the phases $\{\sigma_{\boldsymbol{p}'}\}$ in Eq. (52) can be performed independent of the sign of the Fourier coefficients $W_{\boldsymbol{p}'}$ leading to $\sigma_{\boldsymbol{p}'} = \text{sign}(W_{\boldsymbol{p}'})$. Thus, the same derivation of $\mathcal{F}_{\boldsymbol{w},\boldsymbol{q}'}$ in Eq. (52) can be applied for attractive as well as repulsive interactions. Last but not least, the universal pure functional $\mathcal{F}_{\boldsymbol{w}}$ on the domain $\mathcal{E}_N^1(\boldsymbol{w})$ finally follows as $\mathcal{F}_{\boldsymbol{w}} = \min_{\boldsymbol{q}'} \mathcal{F}_{\boldsymbol{w},\boldsymbol{q}'}$.

### F. Bose-Einstein condensation force

For ground state RDMFT a remarkable property has recently been discovered: the gradient of the universal functional diverges repulsively on the boundary of the allowed regime. This *BEC force* for bosons [3, 4, 6] and *exchange force* for fermions [63] is a consequence of the geometry of quantum states and thus independent of the microscopic properties of the system. In the following, based on the result (43) we confirm the existence of this BEC force also in the context of excited state RDMFT.

This demonstrates that the boundary of the functional's domain $\overline{\mathcal{E}}_N^1(\boldsymbol{w})$ and effectively $\Sigma^\downarrow(\boldsymbol{w}) = \mathrm{spec}^\downarrow(\overline{\mathcal{E}}_N^1(\boldsymbol{w}))$ contains crucial information about the excitation structure of $N$-boson quantum systems.

In the following we therefore consider the boundary of $\Sigma^\downarrow(\boldsymbol{w})$, with a particular emphasis on the neighborhood of the generating vertex $\boldsymbol{v}$ (36). From Eq. (43) we obtain for the derivative of $\overline{\mathcal{F}}_{\boldsymbol{w}}$ with respect to the occupation numbers $\eta_{\boldsymbol{p}'}$ close to the vertex $\boldsymbol{v} = (N - 1 + w, 1 - w, 0, \ldots)$,

$$\frac{\partial \overline{\mathcal{F}}_{\boldsymbol{w}}}{\partial \eta_{\boldsymbol{p}'}}(\boldsymbol{\eta}) \propto \begin{cases} -\frac{1}{\sqrt{\eta_{\boldsymbol{p}'}}} & \text{if } \boldsymbol{p}' \neq \boldsymbol{q}' , \\ -\frac{1}{\sqrt{\eta_{\boldsymbol{p}'} + w - 1}} & \text{if } \boldsymbol{p}' = \boldsymbol{q}' . \end{cases} \tag{53}$$

Here, the momentum $\boldsymbol{q}'$ denotes the momentum corresponding to the first excitation in accordance with Eq. (52). Eq. (53) already reveals that the gradient of $\overline{\mathcal{F}}_{\boldsymbol{w}}$ diverges repulsively whenever one of the occupation numbers $\eta_{\boldsymbol{p}'}$ tends to zero. Consequently, whenever $\boldsymbol{\eta}$ approaches $\boldsymbol{v}$ or any other point on the boundary, the corresponding gradient force is collectively diverging. This BEC force namely contains individual contributions from various polytope facets that are reached. The gradient force is thus indeed collective in the sense that all individual components for each momentum $\boldsymbol{p}'$ diverge. In this context, the zero momentum state requires a separate treatment since we assumed that this state would be macroscopically occupied. To be more specific, our derivation of $\overline{\mathcal{F}}_{\boldsymbol{w}}$ assumed $n_{\boldsymbol{0}} \approx N$ and we used the normalization to substitute $n_{\boldsymbol{0}}$. Hence, the information about $n_{\boldsymbol{0}}$ close to the boundary of $\Sigma^\downarrow(\boldsymbol{w})$ is hidden in all the other occupation numbers. Let us now assume that the upper bound on $n_{\boldsymbol{0}}$ is saturated, i.e., $n_{\boldsymbol{0}} \to N - 1 + w$. Then, for a large dimension $d$ of the one-particle Hilbert space, i.e. a large number of different $\boldsymbol{p}'$, all $N - n_{\boldsymbol{0}}$ bosons not occupying the $\boldsymbol{p} = \boldsymbol{0}$ state can distribute over the remaining orbitals. As a result, either some of the occupation numbers $\eta_{\boldsymbol{p}'}$ are equal to zero or they are all very close to zero. According to Eq. (53) this leads again to a collective repulsive force at the boundary of the domain of $\overline{\mathcal{F}}_{\boldsymbol{w}}$. To elaborate a bit more on these findings, we investigate in the following the divergence of the gradient of $\overline{\mathcal{F}}_{\boldsymbol{w}}$ as a function of the distance

$$D = \frac{1}{N} \sum_{\boldsymbol{p}'} \eta_{\boldsymbol{p}'} - D_0 \tag{54}$$

of a momentum occupation number vector $\boldsymbol{\eta}$ to the vertex $\boldsymbol{v}$ of $\Sigma^\downarrow(\boldsymbol{w})$. Here, $D_0 = (1 - w)/N$ denotes the fraction of non-condensed bosons at the vertex $\boldsymbol{v}$. For this purpose, let us consider a straight path from a starting point $\overline{\boldsymbol{\eta}}$ towards $\boldsymbol{v}$ according to

$$\boldsymbol{\eta}(t) = \overline{\boldsymbol{\eta}} + t(\boldsymbol{v} - \overline{\boldsymbol{\eta}}) \tag{55}$$

with $t \in [0, 1]$. In particular, we obtain for the distance $D$ along this straight path $D(t) = (1 - t)D(0) \equiv (1 - t)\overline{D}$, where $\overline{D}$ denotes the fraction of non-condensed bosons at

the occupation number vector $\overline{\boldsymbol{\eta}}$. Taking the derivative of $\overline{\mathcal{F}}_{\boldsymbol{w}}$ (43) with respect to the distance $D$ defined in Eq. (54) eventually yields after an elementary calculation

$$\frac{\mathrm{d}\overline{\mathcal{F}}_{\boldsymbol{w}}}{\mathrm{d}D}(\boldsymbol{\eta}) \propto -\frac{1}{\sqrt{D}} . \tag{56}$$

Thus, $\mathrm{d}\overline{\mathcal{F}}_{\boldsymbol{w}}/\mathrm{d}D$ diverges as $1/\sqrt{D}$ in the limit $D \to 0$ in analogy to the fermionic exchange force [63] and the BEC force for ground states [3, 6]. Moreover, the corresponding prefactor is always negative and contains all information about the system's specific properties of the interaction.

We thus succeeded in generalizing the concept of a BEC force also to excitations in homogeneous BECs. Motivated by the significance of the energy gap between the ground state and the first excited state for most physical systems, we discussed above the case $r = 2$. Of course, the derivation of the $\boldsymbol{w}$-ensemble universal functional $\overline{\mathcal{F}}_{\boldsymbol{w}}$ and its gradient can be extended in a similar fashion to larger values of $r$ in order to provide access to a larger number of excitation energies. Yet, this would require more mathematical effort which is also due to the degeneracy of the excited states.

## V. SUMMARY AND CONCLUSIONS

To initiate $\boldsymbol{w}$-RDMFT, we derived analytically the universal functionals for the symmetric Hubbard dimer (Sec. III) and the homogeneous Bose gas within the Bogoliubov approximation (Sec. IV) for $r = 2$ non-zero weights $w_j$. These two systems can be seen as ideal starting points for the future development of more sophisticated functional approximations: The Hubbard dimer constitutes the building block of the Hubbard model, one of the most important models in condensed matter physics and the field of ultracold gases. In turn, the Bogoliubov functional represents the bosonic analogue of the pivotal Hartree-Fock functional for fermionic systems [11] since it refers to the regime of small quantum depletion.

Due to the particular suitability of $\boldsymbol{w}$-RDMFT for describing Bose-Einstein condensates and to offer a broad toolbox to the community, we actually derived the functional $\mathcal{F}_{\boldsymbol{w}}$ of the Bose gas in three conceptually different ways. First, we determined $\mathcal{F}_{\boldsymbol{w}}$ as the Legendre-Fenchel transform with respect to the kinetic energy operator of the well-known formula for low-lying excitation energies. From a general point of view, this emphasizes again the scope of functional theories, namely to solve effectively the ground state or excited state problem for an entire class (1) of Hamiltonians of interest. Second, by introducing the concept of $\boldsymbol{w}$-ensemble $v$-representability, we could determine $\mathcal{F}_{\boldsymbol{w}}$ by inverting the map which assigns to kinetic energy operators $\hat{t}$ the respective momentum occupation numbers of a $\boldsymbol{w}$-ensemble state. This approach emphasizes also a severe curse of universality in

functional theories that has not been acknowledged yet: By varying the one-particle terms of the system (e.g., external potential or kinetic energy operator) energy eigenvalues do cross. This in turn leads to a partitioning of the functional's domain into subdomains, each characterized by its own "local" functional. Third, by resorting to the Bogoliubov transformation, we succeeded in executing the Levy-Lieb constrained search and in particular managed to overcome the common phase dilemma. Despite the focus on bosons these three different routes to develop functional approximation as well as the discussion on $\boldsymbol{w}$-ensemble $v$-representability and the curse of universality can be applied to fermions in an analogous manner.

Last but not least, the results for the two systems highlight that the boundary $\partial\overline{\mathcal{E}}_N^1(\boldsymbol{w})$ of the functional's domain $\overline{\mathcal{E}}_N^1(\boldsymbol{w})$ has a particular relevance. To be more specific, the gradient of the universal functional was found to diverge repulsively as the 1RDM approaches $\partial\overline{\mathcal{E}}_N^1(\boldsymbol{w})$. This remarkable result generalizes the recently discovered *exchange force* [63] for fermions and *BEC force* for bosons [3, 4, 6] in their ground states to mixed states. The existence of those forces does not depend on any microscopic details but has a solely geometrical origin. In that sense, these novel concepts emphasize the prominent role that the geometry of reduced quantum states can play in general in advancing functional theories.

## ACKNOWLEDGMENTS

We are grateful to F. Castillo and J.P. Labbé for valuable discussions. We acknowledge financial support from the German Research Foundation (Grant SCHI 1476/1-1) (J.L., C.S.), the Munich Center for Quantum Science and Technology (C.S.) and the International Max Planck Research School for Quantum Science and Technology (IMPRS-QST) (J.L.). The project/research is also part of the Munich Quantum Valley, which is supported by the Bavarian state government with funds from the Hightech Agenda Bayern Plus.

## Appendix A: Derivation of the $\boldsymbol{w}$-ensemble functional in the symmetric Bose-Hubbard dimer

In this section, we derive the $\boldsymbol{w}$-ensemble functional for the symmetric Bose-Hubbard dimer in the total momentum sector with $P = 0$. The allowed values for the discrete momentum $p$ are given by $p_\nu = \pi\nu$ with $\nu = 0, 1$. We denote the operator creating a boson with momentum $p_\nu$ by $\hat{a}_\nu^\dagger$ (see also Sec. III). The two-dimensional subspace $\mathcal{H}_2^{P=0}$ of the two boson Hilbert space $\mathcal{H}_2$ is spanned by the basis states

$$|1\rangle = \frac{1}{\sqrt{2}}(\hat{a}_0^\dagger)^2|0\rangle\,, \tag{A1}$$

$$|2\rangle = \frac{1}{\sqrt{2}}(\hat{a}_1^\dagger)^2|0\rangle\,, \tag{A2}$$

where $|0\rangle$ denotes the vacuum state. In this basis, every two-boson density operator $\hat{\Gamma}$ with spectrum $\boldsymbol{w} = (w_1, 1-w_1)$ can be expressed as

$$\hat{\Gamma}_{\boldsymbol{w}} = \sum_{i,j=1}^2 \Gamma_{ij}^{(\boldsymbol{w})}|i\rangle\langle j|\,. \tag{A3}$$

However, the minimizer states in the GOK variational principle (3) take the form $\hat{\Gamma}_{\boldsymbol{w}} = \sum_{j=1}^2 w_j|\Psi_j\rangle\langle\Psi_j|$, where $|\Psi_j\rangle$ are the eigenstates of the Hamiltonian $\hat{H} = \hat{h} + \hat{W}$. Here, $\hat{W}$ denotes the Hubbard on-site interaction term in Eq. (8). To determine $\Gamma_{ij}^{(\boldsymbol{w})}$ in terms of the weights $w_j$, we expand the eigenstates $|\Psi_j\rangle$ as follows,

$$|\Psi_1\rangle = \alpha_1|1\rangle + \alpha_2|2\rangle\,, \quad |\Psi_2\rangle = \beta_1|1\rangle + \beta_2|2\rangle \tag{A4}$$

with $\alpha_i, \beta_i \in \mathbb{R}$. These expansion coefficients $\alpha_i, \beta_i$ must further satisfy the orthonormality conditions

$$\alpha_1^2 + \alpha_2^2 = 1\,, \tag{A5}$$
$$\beta_1^2 + \beta_2^2 = 1\,, \tag{A6}$$
$$\alpha_1\beta_1 + \alpha_2\beta_2 = 0\,. \tag{A7}$$

Combining Eq. (A3) with Eq. (A4) leads to

$$\Gamma_{11}^{(\boldsymbol{w})} = 1 - \Gamma_{22}^{(\boldsymbol{w})} = w_1\alpha_1^2 + w_2\beta_1^2$$
$$\Gamma_{21}^{(\boldsymbol{w})} = \Gamma_{12}^{(\boldsymbol{w})} = w_1\alpha_1\alpha_2 + w_2\beta_1\beta_2\,. \tag{A8}$$

To express the 1RDM $\hat{\gamma}$ in terms of the matrix elements $\Gamma_{ij}^{(\boldsymbol{w})}$ in the next step, we first recall that $\hat{\gamma}$ is diagonal in momentum representation. Thus, in our case $\hat{\gamma}$ depends only on a single independent parameter due to the normalization $n_0 + n_1 = 2$, where $n_\nu = \text{Tr}_2[\hat{a}_\nu^\dagger\hat{a}_\nu\hat{\Gamma}_{\boldsymbol{w}}]$. Together with Eq. (A3) we arrive at

$$n_0 = \text{Tr}_2[\hat{a}_0^\dagger\hat{a}_0\hat{\Gamma}_{\boldsymbol{w}}] = 2\Gamma_{11}^{(\boldsymbol{w})} = 2\left(w_1\alpha_1^2 + w_2\beta_1^2\right)\,. \tag{A9}$$

Moreover, to determine the $\boldsymbol{w}$-ensemble functional $\mathcal{F}_{\boldsymbol{w}}(\gamma)$, we need to minimize

$$\text{Tr}_2[\hat{W}\hat{\Gamma}_{\boldsymbol{w}}] = U\left(1 + 2\Gamma_{12}^{(\boldsymbol{w})}\right)$$
$$= U\left(1 + 2(w_1\alpha_1\alpha_2 + w_2\beta_1\beta_2)\right) \tag{A10}$$

according to the constrained search formalism with respect to the coefficients $\alpha_i, \beta_i$. Since their three orthonormality conditions together with Eq. (A9) constitute four conditions for four free variables, this minimization can be carried out analytically without much effort. Solving the resulting system of equations (A5), (A6), (A7) and (A9) leads in a straightforward manner to

$$\text{Tr}_2[\hat{W}\hat{\Gamma}_{\boldsymbol{w}}] = U\left(1 \pm \sqrt{(n_0 - 2w_2)(2w_1 - n_0)}\right)\,. \tag{A11}$$

Choosing the minus sign which minimizes the expectation value in Eq. (A10) eventually yields

$$\mathcal{F}_{\boldsymbol{w}}(n_0) = U\left(1 - \sqrt{(n_0 - 2w_2)(2w_1 - n_0)}\right)$$
$$= U\left(1 - \sqrt{n_0(2 - n_0) - 4w_1 w_2}\right), \quad (A12)$$

which is a functional of the momentum occupation number $n_0$ only. Since the functional $\mathcal{F}_{\boldsymbol{w}}$ and its domain are both convex for every $\boldsymbol{w}$ this functional coincides with its relaxed variant $\bar{\mathcal{F}}_{\boldsymbol{w}}$.

Next, we minimize the energy functional $\mathrm{Tr}_1[\hat{\gamma}\hat{t}] + \mathcal{F}_{\boldsymbol{w}}(n_0)$, where $\hat{t} = -t\sum_{\nu=0,1}\cos(\pi\nu)\hat{n}_\nu$, to verify that the result for $E_{\boldsymbol{w}}$ in (3) is in agreement with the eigenergies of $\hat{H} = \hat{t} + \hat{W}$ obtained from an exact diagonalization. The kinetic energy in terms of $n_0$ is given by (recall that $n_1 = 2 - n_0$)

$$\mathrm{Tr}_1[\hat{t}\hat{\gamma}] = -2t(n_0 - 1). \quad (A13)$$

Thus, to calculate $E_{\boldsymbol{w}}$ we need to solve

$$\frac{\partial}{\partial n_0}\left(-2t(n_0 - 1) + \mathcal{F}_{\boldsymbol{w}}(n_0)\right)\bigg|_{n_0 = \tilde{n}_0} = 0 \quad (A14)$$

for the momentum occupation number $\tilde{n}_0$. This leads to

$$\tilde{n}_0 = 1 + \frac{2t(w_1 - w_2)}{\sqrt{4t^2 + U^2}}. \quad (A15)$$

Then, the energy $E_{\boldsymbol{w}}$ follows as

$$E_{\boldsymbol{w}} = -2t(\tilde{n}_0 - 1) + U\left(1 - \sqrt{\tilde{n}_0(2 - \tilde{n}_0) - 4w_1 w_2}\right)$$
$$\equiv w_1 E_1 + w_2 E_2, \quad (A16)$$

where

$$E_1 = U - \sqrt{4t^2 + U^2}, \quad (A17)$$
$$E_2 = U + \sqrt{4t^2 + U^2}. \quad (A18)$$

The Hamiltonian $\hat{H}$ in Eq. (8) in the basis spanned by the states $|1\rangle$ and $|2\rangle$ defined in Eqs. (A1) and (A2), can be represented by the matrix

$$H = \begin{pmatrix} U - 2t & U \\ U & U + 2t \end{pmatrix}, \quad (A19)$$

which has the two eigenvalues $E_1$ and $E_2$ introduced in Eq. (A17) and (A18). Hence, the result for the energy $E_{\boldsymbol{w}}$ in Eq. (A16) is in agreement with the eigenergies obtained from diagonalizing the matrix $H$.

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
