# Peer review of "Deriving density-matrix functionals for excited states"

_SciPost Physics_

## Round 1 · Referee Report · Anonymous (Referee 1) · 2022-12-12

Report
Typos
• Two lines after Eq. (7) "represents" should be changes to "represent".
• In Fig. 1 the legend has $\omega_i$ instead of $w_i$.
• Two lines above Eq. (45) it should be "spanned by the"
Consistency
• The symbol $\equiv$ should also be used in Eq. (4) and (7). Also in Eq. (5) the first equality should probably be \equiv instead of the second one.
• Sometimes the symbol $\text{Tr}_N$ is used (Eq. (3)) and sometime Tr is used (Eq. (2)). It would be better to stick to one.
• Either keep $\hat{\gamma}_{\hat{h}}$ for v-representable 1RDMs or mention that you drop the subscript and that all 1RDMs need to be v-representable. (For example see beginning of Section IV E 2.)
Readability
• The word "dispersion" is confusing and should be removed.
• Above Eq. (2) $\mathcal{E}^N(\mathbf{w})$ should be defined, at least in words.
• For people outside the GOK-field “$r=2$” might not be clear. Please define it, e.g. between brackets.
• In Eq. (35) $\mathcal{S}^d$ is not defined (both $\mathcal{S}$ and $d$).
• The abbreviation GOK should be explained.
• For completeness, add that the weights sum up to one.
Questions/Comments
• In Eq. (A15) why do you not divide out a factor 2 in the fraction?
• All energies from the Hubbard dimer seem to be missing a factor 1/2. For example above Eq. (13) your expression is Tr$_1[\hat{t}\hat{\gamma}]=-4t(n_0-1)$. I would expect $-2t(n_0-1)$. Maybe there is a normalization factor missing.
• Eq. (43) also has an additional factor 1/2 compared to Eq. (25) and Eq. (26). Please check.
• The paragraph after Eq. (10) is a bit lengthy discussion for the fact that the single translation element in the periodic setting is the same operation as the inversion operation. So they are equivalent. Perhaps you could put this when introducing the dimer. Inversion is also a more natural description of the symmetry element than translation + periodicity, so I would rather liked to have seen g(erade) and u(ngerade) designations.
• I got confused about the precise definition of your creation and annihilation operator. Do they keep the standard normalization of the states? And around Eq. (20) is the operator $\beta_0$ not only needed to compensate for the inconvenience of the standard normalization of the states.
• I would like to see the derivation of Eq. (24) from Eq. (19). Maybe in the appendix.
• Section IV C seems a bit off topic at this point in the paper, since it does not seem to be relevant for the functional for the Bogloiubov approximated homogeneous BEC. Perhaps it is better discussed in section V. Further, the state crossing is a well known issue in functional theories e.g. spin-DFT [von Barth and Hedin J Phys C 5, 1629 (1972), Eschrig “The Fundamentals of Density Functional Theory (revised and extended version)” (2003), DFT on graphs [Leeuwen Penz, JCP 155, 244111 (2021)], RDMFT [Giesbertz JCP 143, 054102 (2015)]. The non-analyticity related a lot to the possible kinks in the adiabatic connection mostly discussed in the context of DFT. It would be good to also make connections to that.
• I am not sure about Eq. (41). Eq. (39) and (40) is only for v-representable 1RDMs and the minimum over convex functionals is not necessarily convex. I think you also need Eq. (37) or something more.
• Why has Eq. (45) creation operators and above, the spanning orthonormal states have annihilation operators? That does not seem to be compatible.
• How did you get Eq. (47)?
• After Eq. (50) it is clear what is intended with the statement about $F_{w,q’}$ but the sentence is illogical. It should relate to the biconjugate of $F_{w,q’}$. Perhaps you could just say that you got effectively the same result.
• In Eq. (55) what is the q' referring to?
• After Eq. (55) what do you mean by "gradient force is collectively diverging? Do you mean divergence of all components?
• In Section V, 4th line, $w=0$ is also finite. You probably mean non-zero.
• It would be good to explicitly mention in the title that your work is focused on bosonic systems.
• How general do you expect the BEC force to be?
• It seems that the construction also works for attractive interactions. Is this correct?
Author: Christian Schilling on 2023-01-22 [id 3257]
(in reply to Report 1 on 2022-12-12)
We thank the Referee for this positive assessment of our work and the constructive feedback which helped us to further improve the manuscript.
Typos $\cdot$ Two lines after Eq. (7) "represents" should be changes to "represent". $\cdot$ In Fig. 1 the legend has $\omega_i$ instead of $w_i$ $\cdot$ Two lines above Eq. (45) it should be "spanned by the"
We thank the Referee for pointing out these three typos which we corrected.
Consistency The symbol $\equiv$ should also be used in Eq. (4) and (7). Also in Eq. (5) the first equality should probably be $\equiv$ instead of the second one.
We agree that in old Eqs. (4) and (7)/new Eqs. (5) and (8) ''$\equiv$'' should be used as suggested by the referee. The set of 1RDMs $\overline{\mathcal{E}}^1_N(\boldsymbol{w})$ if defined via the partial trace since it contains all those 1RDMs that are compatible with an $N$-particle state in $\overline{\mathcal{E}}^N(\boldsymbol{w})$. Based on that, $\overline{\mathcal{E}}^1_N(\boldsymbol{w})= \mathrm{conv}(\mathcal{E}^1_N(\boldsymbol{w}))$ is a mathematical consequence of the definition $\overline{\mathcal{E}}^1_N(\boldsymbol{w})\equiv N\mathrm{Tr}_{N-1}(\overline{\mathcal{E}}^N(\boldsymbol{w}))$. To clarify this we interchanged the second and third expression in old Eq. (5)/new Eq. (6).
Sometimes the symbol $\mathrm{Tr}_N$ is used (Eq. (3)) and sometime $\mathrm{Tr}$ is used (Eq. (2)). It would be better to stick to one.
We thank the referee for this comment and stick to $\mathrm{Tr}_N$ to denote the trace over $N$-particles. We therefore added the subscript $N$ in the trace in the paragraph below Eq. (1) and in old Eq. (2)/new Eq. (3).
Either keep $\hat{\gamma}_{\hat{h}}$ for v-representable 1RDMs or mention that you drop the subscript and that all 1RDMs need to be v-representable. (For example see beginning of Section IV E 2.)
The set of $v$-representable 1RDMs consists of all those $\hat{\gamma}$ for which there exists some one-particle Hamiltonian $\hat{h}$ such that $\gamma$ can be obtained via the sequence of maps in old Eq. (29)/new Eq. (30). Thus, given the sequence in old Eq. (29)/new Eq. (30) we can assign a $\hat{h}$ to $\gamma$ and explicitly write $\hat{\gamma}_{\hat{h}}$. However, given the set of $v$-representable 1RDMs it is a priori not known which $\hat{h}$ would correspond to which $\hat{\gamma}$ in this set. In that sense it is not meaningful to label all $v$-representable 1RDMs by $\hat{\gamma}$ by a subscript $\hat{h}$ except for the cases where a fixed $\hat{h}$ is given from the very beginning.
Readability The word "dispersion" is confusing and should be removed.
In the context of optics, a dispersion relation relates the frequency (energy) of a wave to the wave number (or momentum). It is therefore also common in condensed matter physics to talk about a ''dispersion law'' when the energy of a particle and its momentum are related. For instance, $\varepsilon(\boldsymbol{p})=\boldsymbol{p}^2/2m$ is usually referred to as ''free dispersion''. In the same way it is common to refer to a ''dispersion relation'' in the context of the elementary excitations of Bogoliubov quasiparticles (e.g. see textbook [L. Pitaevskii, S. Stringari, Bose-Einstein Condensation and Superfluidity, Oxford University Press (2016)]). To follow standard terminology we therefore feel that it is appropriate to refer to a ''dispersion relation'' for the Bogoliubov excitation spectrum before old Eq. (28)/new Eq. (29) and only removed the term in the context of $t_{\boldsymbol{p}}$.
Above Eq. (2) $\mathcal{E}^N(\boldsymbol{w})$ should be defined, at least in words.
The set $\mathcal{E}^N(\boldsymbol{w})$ is the subset of $N$-particle density operators that satisfy the spectral constraint $\mathrm{spec}^\downarrow(\hat{\Gamma})=\boldsymbol{w}$. To clarify its definition and make it more explicit we added the new Eq. (2) in the manuscript.
For people outside the GOK-field ''$r=2$'' might not be clear. Please define it, e.g. between brackets.
We explain the definition of ''$r=2$'' in the first paragraph of Sec. IV.E. and the corresponding weight vector $\boldsymbol{w}=(w, 1-w, 0, ..)$ is given by the old Eq. (32)/new Eq. (33). To clarify this aspect, we added this information in the first paragraph of Sec. IV.E and added the term ''non-vanishing weights'' also after old Eq. (36)/new Eq. (37).
In Eq. (35) $\mathcal{S}_d$ is not defined (both $\mathcal{S}$ and $d$).
The group $\mathcal{S}_d$ denotes the permutation group of a set of $d$ elements. In our case $d$ is the dimension of the one-particle Hilbert space $\mathcal{H}_1$, i.e. $d=\mathrm{dim}(\mathcal{H}_1)$. We thank the referee for this comment and added the definition of $\mathcal{S}_d$ and $d$ accordingly.
The abbreviation GOK should be explained.
We added the explanation of the abbreviation ''GOK'' in the sentence below Eq. (1).
For completeness, add that the weights sum up to one.
We added this information in the paragraph above the new Eq. (2).
Questions/Comments In Eq. (A15) why do you not divide out a factor 2 in the fraction? All energies from the Hubbard dimer seem to be missing a factor $1/2$. For example above Eq. (13) your expression is $\mathrm{Tr}_1[\hat{t}\hat{\gamma}]=-4t(n_0-1)$. I would expect $-2t(n_0-1)$. Maybe there is a normalization factor missing.
We thank the referee for this comment. Indeed, the Fourier transform of the kinetic energy reads
Eq. (43) also has an additional factor 1/2 compared to Eq. (25) and Eq. (26). Please check.
The factor $1/2$ in old Eq. (43)/new Eq. (44) compared to old Eqs. (25) and (26)/new Eqs. (26) and (27) is due to the introduction of the new variable $\eta_{\boldsymbol{p}^\prime} = n_{\boldsymbol{p}^\prime} + n_{-\boldsymbol{p}^\prime}$ in old Eq. (17)/new Eq.(18). The sum over $\boldsymbol{p}^\prime$ contains each pair $(-\boldsymbol{p}, \boldsymbol{p}), \boldsymbol{p} \neq \boldsymbol{0}$ only once, whereas in the old Eq. (25)/new Eq. (26) the summation is still performed over $\boldsymbol{p}$. To remind the reader of the definition of $\boldsymbol{p}^\prime$ we recall it below the old Eq. (43)/new Eq. (44).
The paragraph after Eq. (10) is a bit lengthy discussion for the fact that the single translation element in the periodic setting is the same operation as the inversion operation. So they are equivalent. Perhaps you could put this when introducing the dimer. Inversion is also a more natural description of the symmetry element than translation + periodicity, so I would rather liked to have seen g(erade) and u(ngerade) designations.
Since both systems the symmetric Bose-Hubbard dimer in Sec. III and the homogeneous BEC in Sec. IV are translational invariant we would like to use Sec. III to already prepare the reader for the more technical Sec. IV. We therefore feel that it is clearer to use momentum occupation numbers in both cases instead of switching to $n_e, n_u$ for the Bose-Hubbard dimer. To improve the readability we followed the suggestion of the referee and moved the paragraph concerning the inversion symmetry to the beginning of Sec. III before we derive the universal functional.
I got confused about the precise definition of your creation and annihilation operator. Do they keep the standard normalization of the states? And around Eq. (20) is the operator $\beta_0$ not only needed to compensate for the inconvenience of the standard normalization of the states.
The operator $\hat{\beta}_0$ in the unitary $\hat{U}$ in old Eq. (20)/new Eq. (21) ensures that the Hamiltonian remains particle-number conserving and that the Bogoliubov quasiparticle vacuum is a state in the $N$-boson Hilbert space. This is both not the case for a standard Bogoliubov transformation violating particle number conservation. Therefore, the purpose of the $\hat{\beta}_0, \hat{\beta}_0^\dagger$ operators is to ensure particle number conservation. We would like to stress here that particle number conservation is absolutely vital in the context of the Levy-Lieb constrained search. Indeed, the universal functional is defined via the minimization over all $N$-particle states mapping to a given 1RDM with fixed normalization $\mathrm{Tr}_1[\hat{\gamma}]=N$. Thus, if we restrict the set of all $N$-particle states (or $\overline{\mathcal{E}}^N(\boldsymbol{w})$) in the Levy-Lieb constrained search to a smaller variational manifold of states in Sec. IV, all states in this variational manifold are $N$-particle states by construction which is ensured by a particle-number conserving Bogoliubov theory. To support the reader we added this information below old Eq. (20)/new Eq. (21).
I would like to see the derivation of Eq. (24) from Eq. (19). Maybe in the appendix.
To keep our paper self-contained we recap in Sec. IV.A the derivation of the grounds state Bogoliubov approximated universal functional discussed in more detail in our previous paper Ref. [3]. Nevertheless, the derivation of old Eq. (24)/new Eq. (25) is already complete up to some straightforward analytical calculations. We start with the expectation value
Section IV C seems a bit off topic at this point in the paper, since it does not seem to be relevant for the functional for the Bogloiubov approximated homogeneous BEC. Perhaps it is better discussed in section V. Further, the state crossing is a well known issue in functional theories e.g. spin-DFT [von Barth and Hedin J Phys C 5, 1629 (1972), Eschrig “The Fundamentals of Density Functional Theory (revised and extended version)” (2003), DFT on graphs [Leeuwen Penz, JCP 155, 244111 (2021)], RDMFT [Giesbertz JCP 143, 054102 (2015)]. The non-analyticity related a lot to the possible kinks in the adiabatic connection mostly discussed in the context of DFT. It would be good to also make connections to that.
The complexity based on state crossings as discussed in Sec. IV. C is indeed relevant for the derivation of excited state functionals in $\boldsymbol{w}$-RDMFT in general and thus also for the Bogoliubov approximated functional. We therefore feel that it is better to discuss this before the derivation of the $\boldsymbol{w}$-ensemble universal functional and also refer back to Sec. IV.C in Sec. IV.E.1.
We thank the referee for providing these interesting references. It is in fact known that crossings of the ground state and first excited state lead to several conceptual issues in both DFT and RDMFT. A problem, which will be even more present in $\boldsymbol{w}$-RDMFT (or GOK-DFT) for $\boldsymbol{w}\neq (1, 0, ...)$. Nevertheless, the references do not discuss the effect of the level crossings on the analyticity of the functional and the splitting of the functional's domain into different cells. To provide the reader with a broader perspective on the consequences of level crossings in functional theories we added in Sec. IV.C a comment with the references suggested by the referee.
I am not sure about Eq. (41). Eq. (39) and (40) is only for v-representable 1RDMs and the minimum over convex functionals is not necessarily convex. I think you also need Eq. (37) or something more.
By referring to the Levy-Lieb constrained search, the universal functional in ground state RDMFT is defined for all 1RDMs and not only pure state $v$-representable 1RDMs as in Gilbert's formulation of RDMFT. The analogous reasoning applies in $\boldsymbol{w}$-ensemble RDMFT. Thus, also the maximization in the Legendre-Fenchel transformation in old Eq. (37)/new Eq. (38) and the minimization in old Eq. (40)/new Eq. (41) over all $\hat{\gamma}\in\overline{\mathcal{E}}^1_N(\boldsymbol{w})$ and not only over $\boldsymbol{w}$-ensemble $v$-representable 1RDMs. This is crucial since we usually do not know a priori which 1RDMs are $\boldsymbol{w}$-ensemble $v$-representable. The idea in Sec. IV.E.1 is to use the knowledge of the energy $E_{\boldsymbol{w}}$ to derive the universal functional $\overline{\mathcal{F}}_{\boldsymbol{w}}$ via a Legendre-Fenchel transformation in old Eq. (37)/new Eq. (38) and the Legendre-Fenchel transform of any function is always convex since the supremum (maximum) over a family of affine functions is convex. In particular, the universal functional can be calculated via the Legendre-Fenchel transform for all $\hat{\gamma}\in\overline{\mathcal{E}}^1_N(\boldsymbol{w}) $ and not only $\boldsymbol{w}$-ensemble $v$-representable 1RDMs as explained in Sec. IV.D.
Why has Eq. (45) creation operators and above, the spanning orthonormal states have annihilation operators? That does not seem to be compatible.
We thank the referee for pointing out that typo which we corrected. The orthonormal states should of course also have creation operators since the quasiparticle operators $\hat{c}_{\boldsymbol{q}}$ annihilate the quasiparticle vacuum.
How did you get Eq. (47)?
Old Eq. (47)/new Eq. (48) is derived in the same way as the expectation value
After Eq. (50) it is clear what is intended with the statement about $\mathcal{F}_{w, q^\prime}$ but the sentence is illogical. It should relate to the biconjugate thereof. Perhaps you could just say that you got effectively the same result.
The universal functional and its biconjugate coincide for all $\boldsymbol{w}$-ensemble $v$-representable 1RDMs. Therefore, the same holds for $\mathcal{F}_{\boldsymbol{w},\boldsymbol{q}^\prime}$, where $\boldsymbol{q}^\prime$ denotes the momentum corresponding to the first excitation.
In Eq. (55) what is the $\boldsymbol{q}^\prime$ referring to?
The momentum $\boldsymbol{q}^\prime$ is defined as the momentum of the lowest-lying quasi-particle excitation on top of the quasiparticle vacuum as defined in old Eq. (45)/new Eq. (46). To remind the reader about this definition we added a respective explanation below old Eq. (55)/new Eq. (56).
After Eq. (55) what do you mean by "gradient force is collectively diverging? Do you mean divergence of all components?
Yes, the gradient force is collective in the sense that all individual components for each momentum $\boldsymbol{p}^\prime$ diverge. We agree that this aspect should be clarified and added this information in the paragraph below old Eq. (55)/new Eq. (56).
In Section V, 4th line, $w=0$ is also finite. You probably mean non-zero.
The referee is right and we corrected the corresponding sentence accordingly.
It would be good to explicitly mention in the title that your work is focused on bosonic systems.
It is worth noticing that the functional for the Hubbard dimer in Sec. III will be the same for the Fermi-Hubbard dimer restricted to the singlet subspace. This is explained in the first paragraph of Sec. III. Moreover, the concepts behind the three approaches to develop universal functional in Sec. IV are equally applicable to fermionic systems. For instance, both Bogoliubov theory for bosons and BCS theory for femions have a similar foundation with respect to the corresponding set of quasifree states (e.g. see [J. Manuceau, A. Verbeure, Commun. Math. Phys.$\mathbf{9}$, 293–302 (1968)], [V. Bach, E.H. Lieb, J.P. Solovej, J. Stat. Phys. $\mathbf{76}$, 3–89 (1994)]). In addition, also the conceptual aspects discussed in Secs. IV.C and IV.D apply to fermions as well as bosons as mentioned in the last sentence before Sec. IV.C. Therefore, despite the focus on bosonic systems, the scope of this paper is not exclusively narrowed to bosons. Thus, we feel that it is indeed appropriate to not specify this further in the title. To further emphasise this point in the paper we added two respective comments in the first paragraph of Sec. IV and at the end of the second paragraph in Sec. V.
How general do you expect the BEC force to be?
This is indeed a very interesting question which still requires further investigations. We expect that for non-zero interactions a universal functional exhibits this diverging gradient at the boundary of its domain. Thus, the BEC force merely originates from the geometry of a certain class of quantum states. However, we would also like to stress that a physical system will only be affected by the BEC force if the minimizer 1RDM is close enough to the boundary of the functional's domain, e.g. when the system is close to a complete BEC. For a general Hamiltonian, in particular for inhomogeneous systems, this regime is rather narrow.
It seems that the construction also works for attractive interactions. Is this correct?
Yes, the same approach works also for universal functional in the case of attractive interactions. In fact, the minimization over the phases $\sigma_{\boldsymbol{p}^\prime}$ in old Eq. (54)/new Eq. (55) can be executed independent of the sign of the Fourier coefficients $W_{\boldsymbol{p}^\prime}$ leading to $\sigma_{\boldsymbol{p}^\prime} = \mathrm{sign}(W_{\boldsymbol{p}^\prime})$. We discussed this aspect in more detail in our previous work [J. Liebert, C. Schilling, Phys. Rev. Research $\mathbf{3}$, 013282 (2021)]. To comment on this we extended the paragraph below old Eq. (54)/new Eq. (55) accordingly.
Author: Christian Schilling on 2023-01-22 [id 3256]
(in reply to Report 2 on 2023-01-02)We thank the referee for the effort made and the positive assessment of our work.
For simplicity let us first discuss the ground state functional which is contained in $\boldsymbol{w}$-RDMFT for $\boldsymbol{w}_0 = (1, 0, ...)$ and comment on general $\boldsymbol{w}$ afterwards. Then, the set $\mathcal{E}^N(\boldsymbol{w}_0)$ of all $N$-particle pure state is not convex and it is well-known that the pure universal functional
The 1RDM is defined as $\hat{\gamma} \equiv N\mathrm{Tr}_{N-1}[\hat{\Gamma}]$, where $\hat{\Gamma}$ is a $N$-particle density operator with normalization $\mathrm{Tr}_N[\hat{\Gamma}]=1$. Due to the normalization $\mathrm{Tr}_1[\hat{\gamma}]=N$ there is no additional factor $N$ needed in the reduction

---

## Round 1 · Referee Report · Anonymous (Referee 2) · 2023-1-2

Strengths
(2) Application to two systems of bosons
(3) Illustration of the universal divergence of the functionals
(4) Review of the w-ensemble 1-RDM theory
Report
Requested changes
(1) Explain in a sentence or two why the functionals are not convex before closure. (2) Verify the third line of Eq. (3) for correct factors of N given the normalization of the 1-RDM to N.

---

## Round 2 · Referee Report · Anonymous (Referee 2) · 2023-2-1

Report

The authors have addressed my comments, and I recommend publication.

---

## Round 2 · Referee Report · Klaas Giesbertz (Referee 3) · 2023-2-13

Report

The authors have made significant improvements to the article. We only have a few points which we would still like to be seen addressed.

  • The factor 2 has now been corrected in the Appendix, but the wrong factor seems still to be present in Eq. (14) and the line before.

  • After eq.(4) 2nd paragraph : “The non-convexity of the set $\mathcal{E}^1_N({\bf w})$ implies that also universal functional $\mathcal{F}_{\bf w}$ is not necessarily convex.” A non-convex domain immediately implies that the functional cannot be convex, so “… also universal functional $\mathcal{F}_{\bf w}$ is not convex”. However, $\mathcal{E}^1_N({\bf w})$ can be convex as the authors explicitly use a convex example themselves, so to say “The non-convexity of the set $\mathcal{E}^1_N({\bf w})$” is not completely correct. Perhaps the sentence could be replaced by something like “Since $\mathcal{E}^1_N({\bf w})$ is not necessarily convex, the functional $\mathcal{F}_{\bf w}$ is not necessarily convex?” Or “In case $\mathcal{E}^1_N({\bf w})$ is not convex, the functional $\mathcal{F}_{\bf w}$ cannot be convex.”?

  • As the minimum over convex functionals is not necessarily convex getting eq.(42) from eq.(40), and eq.(41) is not “immediate”. Consider for example the function $f(x) = \min{x^2, (x-2)^2}$. This is a minimimization over two convex functions, but f(x) is not convex. So using only these arguments, one only has ≤ in eq. (42) or alternatively one could take the convex hull of the current r.h.s.

Nicolas Cartier, Sarina Sutter and Klaas Giesbertz

  • validity: -
  • significance: -
  • originality: -
  • clarity: -
  • formatting: -
  • grammar: -

Author:  Christian Schilling  on 2023-03-02  [id 3424]

(in reply to Report 3 by Klaas Giesbertz on 2023-02-13)

The authors have made significant improvements to the article. We only have a few points which we would still like to be seen addressed.

We highly appreciate the effort made by the referees and reply to these points below.

The factor 2 has now been corrected in the Appendix, but the wrong factor seems still to be present in Eq. (14) and the line before.

We corrected the typos in Eqs. (14) and (15) (Eqs. (13) and (14) in the new manuscript) and the line above Eq. (14).

After eq.(4) 2nd paragraph : ''The non-convexity of the set $\mathcal{E}^1_N(\boldsymbol{w})$ implies that also universal functional $\mathcal{F}$ is not necessarily convex.'' A non-convex domain immediately implies that the functional cannot be convex, so ''… also universal functional Fw is not convex''. However, $\mathcal{E}^1_N(\boldsymbol{w})$ can be convex as the authors explicitly use a convex example themselves, so to say ''The non-convexity of the set $\mathcal{E}^1_N(\boldsymbol{w})$'' is not completely correct. Perhaps the sentence could be replaced by something like ''Since $\mathcal{E}^1_N(\boldsymbol{w})$ is not necessarily convex, the functional $\mathcal{F}$ is not necessarily convex?'' Or ''In case $\mathcal{E}^1_N(\boldsymbol{w})$ is not convex, the functional $\mathcal{F}$ cannot be convex.''?

We understand that point by the referees. Yet, it is not completely uncommon to extend the definition of convexity of a function also to scenarios with non-convex domains (see, e.g., 'Convex functions on non-convex domains', Econmics Letters 22, 251-255, 1986). Nonetheless, as a compromise we tweaked the corresponding paragraph, below Eq. (4).

As the minimum over convex functionals is not necessarily convex getting eq.(42) from eq.(40), and eq.(41) is not ''immediate''. Consider for example the function $f(x)=\min(x^2,(x-2)^2)$. This is a minimization over two convex functions, but $f(x)$ is not convex. So using only these arguments, one only has $\leq$ in eq. (42) or alternatively one could take the convex hull of the current r.h.s.

We thank the referees for this important comment and we added a lower convex envelop operation `conv' on the rhs of Eq.(40) in the revised manuscript. It actually even turns out that the minimum of the family of functions defined in new Eq. (43) is not convex. Because of this, we had to modify some statements in the subsequent sections (paragraph below Eq. (43), last paragraph of Sec. IV.E.2, first paragraph of Sec. IV.E.3 and the end of the second paragraph of Sec. V).

---

## Round 2 · List of Changes

1) We corrected the three typos pointed out by referee 1.

2) We interchanged the second and third expression in old Eq. (5)/new Eq. (6).

3) We added the subscript $N$ in the trace in the paragraph below Eq. (1) and in old Eq. (2)/new Eq. (3).

4) We removed the term ''dispersion relation'' in the context of $t_{\boldsymbol{p}}$ below old Eq. (15)/new Eq.(16) and added the word ''relation'' to '' dispersion relation'' below old Eq. (28)/new Eq. (29).

5) We added the definition of the set $\mathcal{E}^N(\boldsymbol{w})$ as a new Eq. (2).

6) We added the term ''non-vanishing weights'' in the first paragraph of Sec. IV.E. also after old Eq. (36)/new Eq. (37) to explain the definition of $r=2$.

7) We added the definition of the set $\mathcal{S}_d$ and the dimension $d$ in the sentence below old Eq. (35)/new Eq. (36).

8) We added the explanation of the abbreviation ''GOK'' in the sentence below Eq. (1).

9) We added the normalization of the weight vector, $\sum_{i=1}^D w_i=1$, in the paragraph above the new Eq.(2).

10) There was a wrong factor of $2$ in the Fourier transform of the kinetic energy in Appendix A and we corrected Eqs. (A13)-(A19) accordingly.

11) We recall the definition of $\boldsymbol{p}^\prime$ below the old Eq. (43)/new Eq. (44).

12) We added below old Eq. (20)/new Eq. (21) an explanation of the purpose of the operators $\hat{\beta}_0, \hat{\beta}^\dagger_0$ to stress that they ensure particle-number conservation.

13) We added below old Eq. (22)/new Eq. (23) an additional explanation how old Eq. (24)/new Eq. (25) is derived and refer the reader to Ref. [51].

14) We added a further comment regarding consequences of level crossings in functional theories in Sec. IV.C and added the references [54-56].

15) We corrected a typo in old Eq. (45)/new Eq. (46).

16) We added an additional explanation how old Eq. (47)/new Eq. (48) is derived below old Eq. (47)/new Eq. (48).

17) We recall the definition of $\boldsymbol{q}^\prime$ below old Eq. (55)/new Eq. (56).

18) We added below Eq. (55)/new Eq. (56) an explanation of the term ''gradient force is collectively diverging'' in the context of the BEC force.

19) In Sec. V we replaced ''$r=2$ finite weights'' by ''$r=2$ non-zero weights''.

20) We added in the first paragraph of Sec. IV a sentence explaining that the three approaches to derive a universal functional can be applied to fermions on an equal footing and comment on the Legendre-Fenchel transform as an explicit example. We also added a comment on the validity of these different routes to derive universal functionals and of Sec. IV.C and IV. D in the last sentence of the second paragraph in Sec. V.

21) We added a comment on attractive interactions below Eq. (54)/new Eq. (55).

22) We added an explanation why the universal functional $\mathcal{F}_{\boldsymbol{w}}$ is in general not convex before applying an exact convex relaxation in the paragraph below old Eq. (2)/new Eq. (3).

23) We added an explanation that we use for simplicity the same symbol for the one-particle Hamiltonian $\hat{h}$ on the $N$-particle Hilbert space and the one-particle Hilbert space below old Eq. (2)/new Eq. (3).

---

## Editorial Decision

resubmitted